# Protein quality control in the nucleolus safeguards recovery of epigenetic regulators after heat shock

Maria Azkanaz[1], Aida Rodríguez López[1], Bauke de Boer[1], Wouter Huiting[2], Pierre-Olivier Angrand[3], Edo Vellenga[1], Harm H Kampinga[2], Steven Bergink[2], Joost HA Martens[4], Jan Jacob Schuringa[1†], Vincent van den Boom[1†]*

[1]Department of Experimental Hematology, Cancer Research Center Groningen, University Medical Center Groningen, University of Groningen, Groningen, Netherlands; [2]Department of Biomedical Sciences of Cells and Systems, University Medical Center Groningen, University of Groningen, Groningen, Netherlands; [3]Cell Plasticity & Cancer, Inserm U908, University of Lille, Lille, France; [4]Department of Molecular Biology, Faculty of Science and Medicine, Radboud Institute for Molecular Life Sciences, Radboud University Nijmegen, Nijmegen, Netherlands

**Abstract** Maintenance of epigenetic modifiers is of utmost importance to preserve the epigenome and consequently appropriate cellular functioning. Here, we analyzed Polycomb group protein (PcG) complex integrity in response to heat shock (HS). Upon HS, various Polycomb Repressive Complex (PRC)1 and PRC2 subunits, including CBX proteins, but also other chromatin regulators, are found to accumulate in the nucleolus. In parallel, binding of PRC1/2 to target genes is strongly reduced, coinciding with a dramatic loss of H2AK119ub and H3K27me3 marks. Nucleolar-accumulated CBX proteins are immobile, but remarkably both CBX protein accumulation and loss of PRC1/2 epigenetic marks are reversible. This post-heat shock recovery of pan-nuclear CBX protein localization and reinstallation of epigenetic marks is HSP70 dependent. Our findings demonstrate that the nucleolus is an essential protein quality control center, which is indispensable for recovery of epigenetic regulators and maintenance of the epigenome after heat shock.
DOI: https://doi.org/10.7554/eLife.45205.001

*For correspondence:
v.van.den.boom@umcg.nl

[†]These authors contributed equally to this work

Competing interests: The authors declare that no competing interests exist.

## Introduction

The epigenetic landscape of a cell is fundamentally important for various DNA metabolic processes including gene transcription, DNA replication and DNA repair (*Tessarz and Kouzarides, 2014*). Proper maintenance of the epigenome is essential for cell viability, and increasing evidence suggests that changes in the chromatin landscape are causally related to aging-associated functional decline of a cell and ultimately cell death (*Booth and Brunet, 2016*). Maintenance of the epigenetic landscape can only be guaranteed by correct positioning and activity of epigenetic modifiers across the genome, and may be threatened by proteotoxic stress. Importantly, misregulation of epigenetic modifiers (i.e. mutations, misexpression) is frequently observed in various cancer types, underlining that regulation of the epigenetic landscape is essential for appropriate cellular functioning (*Dawson and Kouzarides, 2012*).

The Polycomb group (PcG) protein family of epigenetic modifiers warrants proper regulation of stem cell self-renewal and cell lineage specification. PcG proteins reside in the canonical Polycomb repressive complexes 1 (PRC1) and 2 (PRC2) (*Simon and Kingston, 2013*). The PRC2 complex contains EZH1/2-dependent methyltransferase activity toward histone H3 at lysine 27 (H3K27me3) (*Cao et al., 2002*; *Ezhkova et al., 2011*; *Kirmizis et al., 2004*; *Kuzmichev et al., 2002*; *Shen et al.,*

**eLife digest** All cells in our bodies contain the same sequence of DNA, hence the same genes, in a compartment called the nucleus. Yet different sets of genes are switched on in different types of cells. Cells achieve this by a process called epigenetic regulation. Proteins known as epigenetic regulators modify DNA and its associated proteins in ways that can turn genes on or off. Different types of cells contain different epigenetic regulators, and so express different genes. The Polycomb group proteins (or PcG for short) turn their target genes off and are important to maintain the identity of a cell. When the target genes of PcG proteins are inadvertently switched on, this may lead to changes in the fate of cells, potentially resulting in diseases such as cancer. So, it is important that cells keep the PcG proteins active where necessary, even in the face of stress.

Cellular stresses come in several forms but often interfere with the normal activities of proteins. If cells experience high temperatures, they can experience a stress known as heat shock. This can cause proteins, including PcG proteins, to unfold. Azkanaz et al. have now investigated what happens to PcG proteins in cells experiencing heat shock, and how these cells try to limit the damage this causes.

Azkanaz et al. conducted their experiments on healthy and cancerous human blood cells. After exposing the cells to half an hour of high temperature the PcG proteins disappeared from the genes they were switching off. This means that cells exposed to heat shock lose their epigenetic control machinery, which may lead to permanent changes to epigenetic modifications found across the genome when not quickly reinstalled. PcG proteins, and another group of proteins called the heat shock proteins, were found to move to a compartment within the nucleus called the nucleolus. While the cells had returned to body temperature and were recovering from the heat shock, the heat shock proteins helped the PcG proteins fold back into their proper shapes. The PcG proteins then left the nucleolus and returned to their target genes, where they reinstalled the epigenetic marks.

These experiments show that heat shock causes a temporary loss of epigenetic regulators from their target genes and that the nucleolus acts as a protein quality control center. Future experiments might explore how PcG proteins get to the nucleolus after heat shock and how impaired protein quality control (i.e. upon aging) may lead to alterations of the epigenetic landscape in a cell. Deeper knowledge of this process could help us to understand how cells can recover from stress.

DOI: https://doi.org/10.7554/eLife.45205.002

2008). PRC1 can ubiquitinate histone H2A at lysine 119 (H2AK119), by means of its E3 ligase subunit RING1A/B (*de Napoles et al., 2004*; *Wang et al., 2004*). PRC1 and PRC2 frequently colocalize at target genes and initially a hierarchical model was proposed for PRC1/2 function, where the CBX subunit of PRC1 recognizes the PRC2 mark H3K27me3, placing PRC1 function downstream of PRC2. However, recent studies have shown that PRC1 and PRC2 recruitment to chromatin, and associated histone modifying activities, can also be independent of each other (for a review see *Blackledge et al., 2015*).

Alternatively, non-canonical PRC1-deposited H2AK119ub was shown to independently sequester PRC2 complexes. Work from many labs, including ours, has underlined the importance of (non-)canonical PRC1 complexes for regulation of cellular identity of normal hematopoietic stem cells and leukemic stem cells (*Iwama et al., 2004*; *Lessard and Sauvageau, 2003*; *Park et al., 2003*; *Rizo et al., 2008*; *Rizo et al., 2010*; *Rizo et al., 2009*; *van den Boom et al., 2016*; *van den Boom et al., 2013*). It is therefore evident that preservation of Polycomb-mediated epigenetic regulation is essential to maintain cell identity and prevent cellular transformation and that, in case of cellular stress induced proteotoxicity, the functionality of the epigenetic machinery is guaranteed.

In this study, we investigated the stability of the epigenetic machinery in response to heat shock (HS). HS is known to lead to a general shutdown of transcription. Cellular stressors like HS and proteasome inhibition induce a quick depletion of the free ubiquitin pool and this coincides with a quick reduction of ubiquitinated histone H2A (H2AK119ub) in the cell (*Carlson and Rechsteiner, 1987*; *Dantuma et al., 2006*; *Mimnaugh et al., 1997*). These data suggest that the epigenome may be affected by HS. A study in *Drosophila* cells indeed showed that HS leads to dramatic alterations of the 3D chromatin architecture as a consequence of weakening insulators between topologically

associating domains (TADs) and newly formed architectural protein binding sites (*Li et al., 2015*). In addition, Polycomb complexes were redistributed to active promoters/enhancers and formed inter-TAD interactions, likely resulting in transcriptional silencing.

For a subset of genes, however, in particular the genes encoding the heat-shock proteins (HSPs), HS does not cause a decrease but rather an increase in gene transcription. This response is referred to as the Heat Shock Response and mediated largely by the so-called Heat Shock Transcription factor-1 (HSF-1) (*Akerfelt et al., 2010*). HSPs function as molecular chaperones, not only guiding co-translational folding under normal conditions but also serving to refold heat-unfolded proteins. If proteins cannot be correctly refolded, they can be poly-ubiquitinated and degraded by the proteasome. Importantly, the intracellular pool of 'free' ubiquitin that is used for poly-ubiquitination of proteins is limited (*Carlson and Rechsteiner, 1987*). As such, HSPs prevent protein dysfunction and aggregation, a hallmark of various age-related neurodegenerative diseases like Alzheimer's disease and Parkinson's disease (*Hartl et al., 2011*; *Kampinga and Berglink, 2016*; *Morimoto, 2008*).

In this study, we specifically investigated the effects of HS on the epigenetic machinery and how this is restored upon return to physiological temperatures. We observed that PRC1 and PRC2 subunits and various other chromatin modifiers accumulate in the nucleolus upon HS. Various labs have reported on reversible accumulation of reporter-proteins in the nucleus upon heat shock (*Miller et al., 2015*; *Nollen et al., 2001*; *Park et al., 2013*), but whether this also holds true for endogenous proteins, and what could be the physiological relevance of this process, has remained unclear. We find that the nucleolar accumulation of these epigenetic regulators coincides with a displacement of PRC1 and PRC2 from their target genes and a dramatic loss of H2AK119ub and H3K27me3. Most importantly, the nucleolar accumulation is reversible in an HSP70-dependent manner allowing epigenetic recovery. Our data demonstrate that the nucleolus is an essential protein quality control (PQC) center that serves to restore the epigenomic landscape after conditions of proteotoxic stress in an HSP-dependent manner.

## Results

### Heat shock induces nucleolar localization of CBX proteins

To investigate the effects of thermal stress on the epigenetic machinery, we analyzed the localization of PRC1 subunits in response to heat shock (HS). We transduced cord blood CD34[+] stem/progenitor cells using a GFP-CBX2 lentiviral vector (*Figure 1A*). Importantly, GFP pull out experiments in K562 GFP-CBX2 cells confirmed that GFP-CBX2 was properly incorporated in the PRC1 complex (*Figure 1—figure supplement 1A*) and ChIP-seq experiments in K562 GFP-CBX2 and K562 wild-type cells showed that GFP-CBX2 target genes largely overlapped with endogenous CBX2 target genes (based on ENCODE/Broad Institute data) and H2AK119ub enriched genes (*van den Boom et al., 2016*). These data underline that the GFP-CBX2 fusion protein is incorporated into a fully functional PRC1 complex. Next, we studied the localization of GFP-CBX2 in untreated and heat shocked cord blood CD34[+] cells. Whereas GFP-CBX2 was homogenously distributed throughout the nucleus in untreated cells, cells that received a HS (30 min, 44°C) displayed strong accumulations of GFP-CBX2 in subnuclear domains, both in cells that were fixed after HS (*Figure 1B*) and living cells (*Figure 1C*). Similarly, K562 leukemic cells also showed HS-induced relocalization of GFP-CBX2 to subnuclear domains (*Figure 1—figure supplement 2A*). Transmission images suggested that HS induces relocalization of GFP-CBX2 to nucleoli, which was confirmed by immunofluorescence analyses using antibodies against the nucleolar proteins NPM1 and Fibrillarin (*Figure 1D* and *Figure 1—figure supplement 2B–D*). GFP-CBX2 localized directly around Fibrillarin, which is confined to the dense fibrillar component (DFC) of the nucleolus, and partially colocalized with the granular component (GC) protein NPM1. Taken together, these data suggest that GFP-CBX2 is most enriched in the granular component of the nucleolus (*Boisvert et al., 2007*). The kinetics of HS-induced nucleolar accumulation of GFP-CBX2 were both dependent on the duration and temperature of the HS (*Figure 1E* and *Figure 1—figure supplement 2E–F*). Cells exposed to a temperature of 42°C also displayed nucleolar localization of GFP-CBX2 albeit with slower kinetics.

To investigate whether HS-induced nucleolar relocalization is common to all PRC1-associated CBX paralogs, K562 cell lines were generated expressing GFP-CBX4, GFP-CBX6, GFP-CBX7 or GFP-CBX8. Indeed, all these CBX paralogs displayed HS-induced nucleolar accumulation (*Figure 1F and*

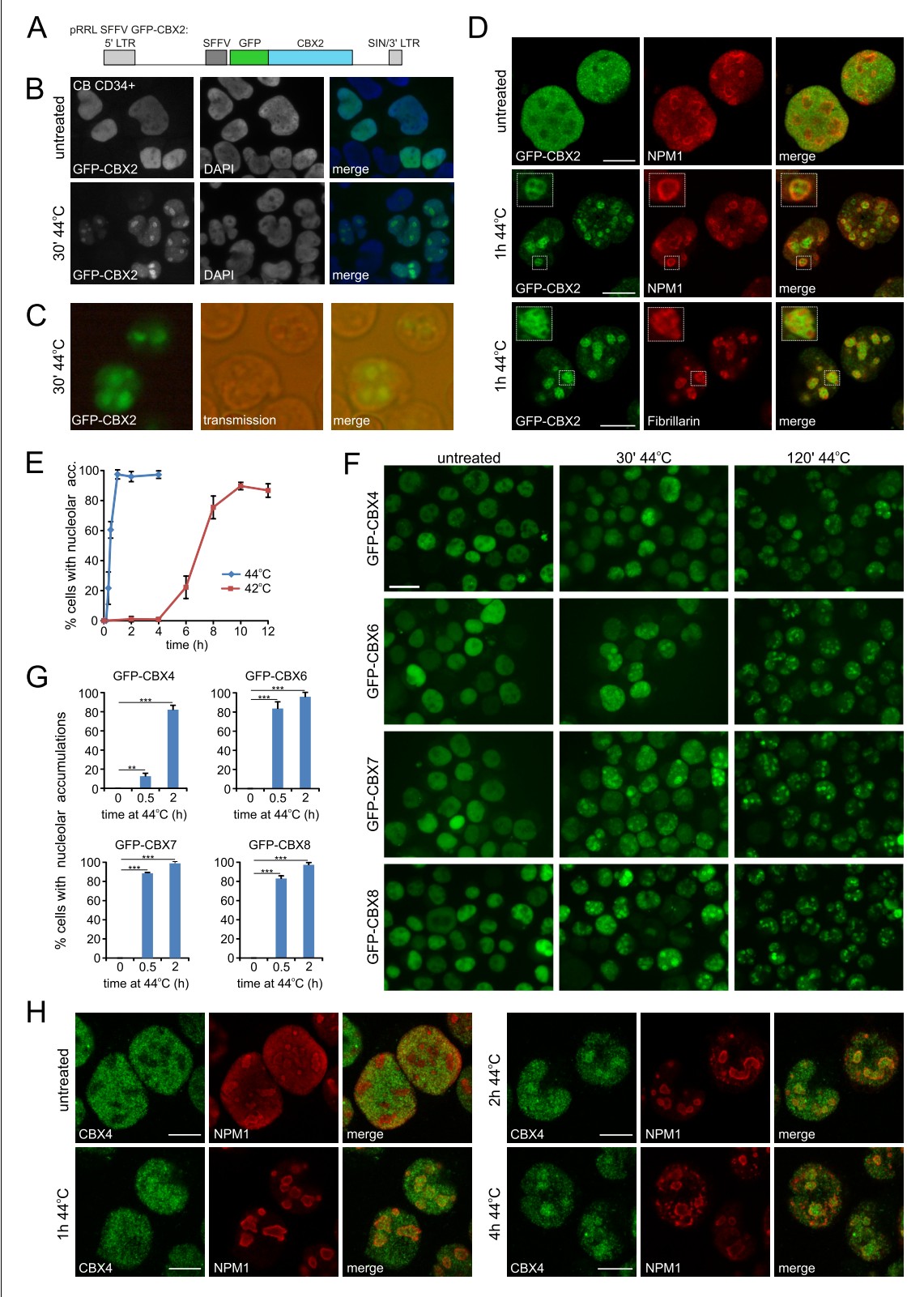

**Figure 1.** Heat shock induces nucleolar relocalization of CBX proteins. (**A**) Graphical representation of the pRRL SFFV GFP-CBX2 lentiviral vector that was used in this study. (**B**) GFP-CBX2 localization in fixed untreated or heat shocked (30 min, 44°C) cord blood (CB) CD34+ cells. (**C**) GFP-CBX2 localization in live K562 GFP-CBX2 cells directly after HS. (**D**) Confocal images of untreated and heat shocked K562 GFP-CBX2 cells that were fixed and stained with either anti-NPM1 or anti-Fibrillarin antibodies. Scale bar represents 10 μm. (**E**) Percentage of cells with nucleolar accumulation of GFP-

*Figure 1 continued on next page*

*Figure 1 continued*

CBX2 at continuous exposure at 42˚C or 44˚C. Error bars indicate the mean ± SD calculated from independent microscopical images (n = 4; total cell number 70–220). Similar results were obtained in independent experiments. (F) GFP-CBX4, GFP-CBX6, GFP-CBX7 and GFP-CBX8 localization in K562 cells in untreated or heat shocked (30 min, 44˚C) cells. Scale bar represents 25 μm. (G) Quantification of percentage of cells with nucleolar accumulations in designated K562 cell lines after HS (30 min or 2 hr, 44˚C). Error bars indicate the mean ± SD calculated from independent microscopical images (n = 3; total cell number 70–100). Statistical analysis was performed using Student's t-test, **p<0.01 and ***p<0.001. (H) Confocal images of untreated and heat shocked K562 cells that were fixed and stained with anti-CBX4 and anti-NPM1. Scale bar represents 10 μm.

DOI: https://doi.org/10.7554/eLife.45205.003

The following figure supplements are available for figure 1:

**Figure supplement 1.** GFP-CBX2 is incorporated in the PRC1 complex and shows overlapping chromatin binding compared to endogenous CBX2 and H2AK119ub.

DOI: https://doi.org/10.7554/eLife.45205.004

**Figure supplement 2.** Heat-shock-induced GFP-CBX2 nucleolar localization kinetics are temperature dependent.

DOI: https://doi.org/10.7554/eLife.45205.005

**Figure supplement 3.** The N-terminus of CBX2 and CBX8 including the chomo domain is sufficient for HS-induced nucleolar localization.

DOI: https://doi.org/10.7554/eLife.45205.006

*G*). Importantly, using immunofluorescence, we also observed nucleolar accumulation of endogenous CBX4 upon HS, both in K562 and HL60 cells (*Figure 1H* and *Figure 1—figure supplement 2G*). Based on these data, we hypothesized that a common domain in these proteins is sufficient for HS-induced nucleolar accumulation. Since the homology between these various CBX proteins is confined to the chromodomain (N-terminus) and Pc box (C-terminus), we generated GFP-CBX2 and GFP-CBX8 fusions only containing the chromobox (GFP-CBX2 [2-63]) or the chromobox and AT hook (GFP-CBX2 [2-96]) (*Figure 1—figure supplement 3A*). Strikingly, all generated truncated GFP-CBX fusion proteins displayed nucleolar localization after HS, suggesting the presence of the chromodomain is sufficient to induce nucleolar localization after HS (*Figure 1—figure supplement 3B*). The kinetics of HS-induced relocalization of truncated CBX proteins were slightly slower, suggesting that also non-homologous peptide stretches in CBX proteins contribute to nucleolar relocalization.

## HS induces large-scale changes in the nucleolar proteome

To verify the HS-induced accumulation of PcG proteins in the nucleolus, we isolated nucleoli from heat shocked and untreated GFP-CBX8 expressing K562 cells (*Figure 2A*). Microscopic analysis of unfixed isolated nucleoli followed by image analysis showed a robust increase in GFP-CBX8 signal in nucleoli isolated for cells directly after HS (*Figure 2B and C*). This observation was confirmed by fixing nucleoli and subsequent counterstaining with DAPI (*Figure 2D*). Next, we performed western blot analysis on isolated cytoplasmic, nucleoplasmic and nucleoli fractions from GFP-CBX8 cells, which confirmed the presence of the nucleolar marker Fibrillarin in the nucleoli fraction and clearly showed an increase of GFP-CBX8 in the nucleolar fraction after HS (*Figure 2E*). To analyze changes in the localization of endogenous PRC1 subunits, we isolated nucleoli from wild-type K562 cells and similarly performed western analysis of the cytoplasmic, nucleoplasmic and nucleoli fractions (*Figure 2F*). Also here Fibrillarin was prominently found in the nucleolar fraction and beta-actin was confined to the cytoplasmic fraction. Clearly, both endogenous CBX4 and CBX8 were enriched in the nucleolar fraction after HS. In addition, also endogenous RING1B levels were slightly elevated in the nucleolar fraction after HS. To validate these results in an independent cell line, we analyzed cellular fractions isolated from HL60 cells (*Figure 2G*). Also here, we observed a robust shift of endogenous CBX4, CBX8 and RING1B to the nucleolus after HS.

What could be the physiological relevance of such a shift of proteins regulating DNA-dependent processes to the nucleolus? It has been known that one of the most dramatic morphological changes in heat treated nuclei is the swelling of nucleoli (*Welch and Suhan, 1986*). Whereas initially considered as heat-induced damage, several lines of independent observations have suggested that this might rather reflect a regulated, HSP-dependent process in which the nucleolus serves as a temporal storage site for unfolded proteins during proteotoxic stress (*Nollen et al., 2001*; *Ohtsuka et al., 1986*; *Welch and Feramisco, 1984*). In line with this hypothesis, we also found that both HSP70 and DNAJB1 are accumulating in the nucleolus after HS (*Figure 2G*), which is in agreement with earlier

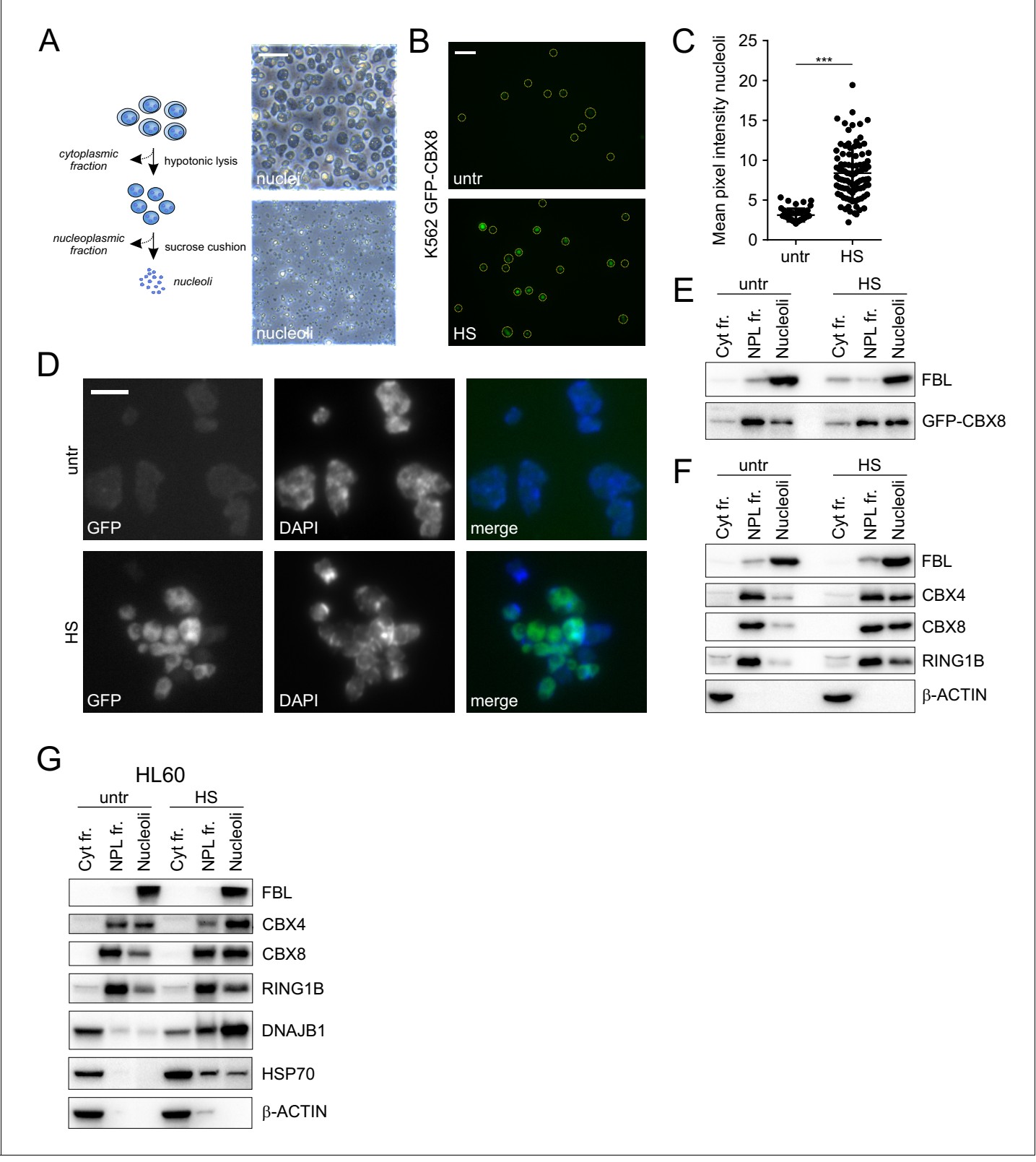

**Figure 2.** Cellular fractionation shows HS-induced nucleolar relocalization of endogenous Polycomb proteins. (**A**) Graphical representation of isolation of cytoplasmic, nucleoplasmic and nucleoli fractions and transmission images of isolated nuclei and nucleoli. Scale bar represents 40 µm. (**B**) Representative images of non-fixed nucleoli isolated from untreated or heat-shocked (1 hr, 44°C) K562 GFP-CBX8 cells. Scale bar represents 10 µm. (**C**) Quantification of the GFP-CBX8 fluorescent signal of nucleoli isolated form untreated (n = 47) or heat-shocked (n = 92) cells. Error bars indicate

*Figure 2 continued on next page*

*Figure 2 continued*

mean ± SEM. Statistical analysis was performed using Student's t-test, ***p<0.001. (D) Representative image from fixed nucleoli isolated from untreated or heat-shocked (1 hr, 44°C) K562 GFP-CBX8 cells and counterstained with DAPI. Scale bar represents 5 μm. (E) Western blot analyses of cytoplasmic, nucleoplasmic and nucleoli fractions from untreated and heat-shocked K562 GFP-CBX8 cells stained with anti-Fibrillarin (FBL) and anti-GFP antibodies. (F) Western blot analyses of cytoplasmic, nucleoplasmic and nucleoli fractions from untreated and heat-shocked K562 cells stained with anti-Fibrillarin (FBL), anti-CBX4, anti-CBX8, anti-RING1B and anti-β-ACTIN antibodies. (G) Western blot analyses of cytoplasmic, nucleoplasmic and nucleoli fractions from untreated and heat-shocked HL60 cells stained with anti-Fibrillarin (FBL), anti-CBX4, anti-CBX8, anti-RING1B, anti-DNAJB1, anti-HSP70 and anti-β-ACTIN antibodies.

DOI: https://doi.org/10.7554/eLife.45205.007

observations that DNAJB1 and HSP70s can translocate to the nucleolus after HS (*Ohtsuka et al., 1986*; *Welch and Feramisco, 1984*).

Next, we performed label-free quantification on the nucleolar proteome in untreated and heat shocked K562 cells by LC/MS-MS analyses. In total, we identified 1279 proteins, and the nucleolar proteins NPM1 and Fibrillarin were among the most abundant proteins (*Figure 3—figure supplement 1A*). GO term analysis of the top ten percent identified proteins based of LFQ values showed strong enrichment for GO terms related to ribosome biogenesis and rRNA processing (*Figure 3—figure supplement 1B*). Subsequently, we analyzed proteins that were up or down in the nucleolus after HS. MaxQuant-based label-free quantification of two independent experiments measured in triplicate resulted in the identification of 153 significantly enriched proteins in the nucleolus after HS and six depleted proteins (*Figure 3A*, *Figure 3—figure supplement 2A*, *Supplementary files 1–2*). Nucleolar proteins like Fibrillarin and NPM1 were not affected by HS (*Figure 3—figure supplement 2A*). Interestingly, proteins enriched in the nucleolus after HS associated with GO terms related to chromatin modification, gene expression, DNA repair, histone ubiquitination and protein refolding (*Figure 3B*). Consistent with our IF and biochemical fractionation data, various PRC1 subunits, including CBX2, RING1A, RING1B and PHC2 were significantly enriched in the nucleolus upon HS (*Figure 3C* and *Figure 3—figure supplement 2B*). Intriguingly, we also found that PRC2 subunits EZH2, SUZ12 and EED were enriched (*Figure 3C* and *Figure 3—figure supplement 2B*) and this HS-induced nucleolar accumulation of EZH2 and SUZ12 was confirmed using western blot analysis on nucleolar fractions isolated from K562 or HL60 cells (*Figure 3D*). HS-induced nucleolar accumulation of EZH2 was independently confirmed using immunofluorescence analyses in K562 cells, HL60 cells, and primary non-transformed CD34+mobilized peripheral blood stem cells (mPBSCs) (*Figure 3—figure supplement 3A–C*). Intra-nucleolar levels of H3K27me3 and H2AK119ub were not increased in heat shocked cells versus untreated cells, suggesting that PcG proteins are not involved in Polycomb-mediated silencing of nucleolar chromatin (*Figure 3E*). In addition to these PcG proteins, many other chromatin and transcription regulating proteins were found to be enriched in the nucleolus after HS, including members of the chromodomain helicase DNA-binding (CHD) family and the FACT chromatin remodeling complex that both can remodel chromatin (*Marfella and Imbalzano, 2007*; *Winkler and Luger, 2011*), and the PAF complex, which regulates release of RNAPII into processive elongation (*Van Oss et al., 2017*) (*Figure 3F* and *Figure 3—figure supplement 2C*). In addition, accumulation of BRD proteins and JMJD6 was observed in the nucleolus after HS (*Figure 3G* and *Figure 3—figure supplement 2D*). BRD4 and JMJD6 are co-bound to enhancers and regulate promoter-proximal pause-release of RNAPII (*Liu et al., 2013*). Taken together, these data show that HS induces strong shifts of various chromatin and transcription regulators toward the nucleolus.

In line with our western analysis of cellular fractions, our proteomic analyses showed several chaperone proteins to be enriched in the nucleolus after HS. These included members of the HSP70 chaperone family (HSPA1A/B), DNAJB1, DNAJC7 and the small heat shock protein HSPB1 (*Figure 3H* and *Figure 3—figure supplement 2E*). Strikingly, many other chaperones, including members of the HSP90 family were not or weakly enriched in the nucleolus after HS showing the response is specific and suggesting that this subset of HSPs may somehow have functional implications in this response. In addition to the HSPs, we also observed a strong accumulation of 26S proteasome subunits in the nucleolus after HS (*Figure 3I* and *Figure 3—figure supplement 2F*). It is possible that post-HS proteasomal degradation of damaged proteins occurs in the nucleolus. This is

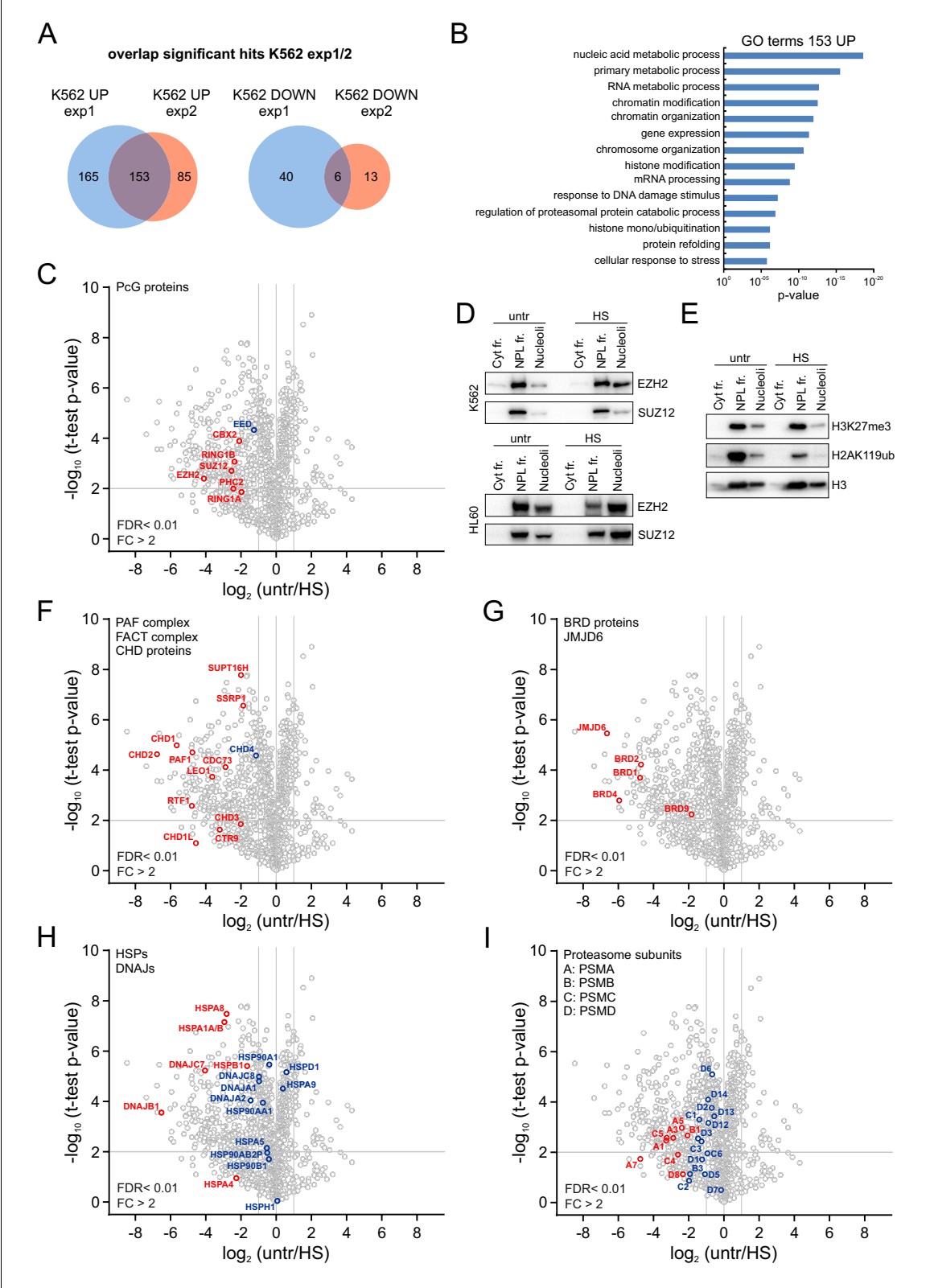

**Figure 3.** Heat shock induces nucleolar accumulation of Polycomb proteins, chromatin regulators and heat-shock proteins. (**A**) Venn diagrams showing overlap of significantly enriched/depleted proteins in nucleoli after HS (1 hr, 44°C) as identified in two independent experiments. Nucleolar fractionations were performed on K562 cells (untreated, HS) and samples were analyzed using liquid chromatography-tandem mass spectrometry (LC-MS/MS) in triplicates, followed by data analysis using MaxQuant and Perseus software. (**B**) Gene ontology (GO) analysis of overlapping proteins that

*Figure 3 continued on next page*

*Figure 3 continued*

were significantly enriched in the nucleolus after HS. (**C**) Volcano plot showing nucleolar proteins in untreated and heat shocked K562 cells and highlighting enriched PRC1 and PRC2 subunits. Statistical analysis was performed using Student's t-test (false discovery rate (FDR) < 0.01; fold change (FC) > 2). Significantly changed proteins are marked in red. (**D**) Western blot analyses of cytoplasmic, nucleoplasmic and nucleoli fractions from untreated and heat-shocked K562 and HL60 cells stained with anti-EZH2 and anti-SUZ12 antibodies. (**E**) Western blot analyses of cytoplasmic, nucleoplasmic and nucleoli fractions from untreated and heat-shocked K562 cells stained with antibodies directed against histone H3, H3K27me3 and H2AK119ub. (**F**) Volcano plot showing (significantly) enriched subunits of the PAF and FACT complex and CHD proteins. (**G**) Volcano plot displaying significantly enriched BRD family members and the JMJD6 protein. (**H**) Volcano plot showing (significantly) enriched HSP70 and DNAJ heat shock proteins. (**I**) Volcano plot highlighting all (significantly) enriched proteasomal subunits.

DOI: https://doi.org/10.7554/eLife.45205.008

The following figure supplements are available for figure 3:

**Figure supplement 1.** LC-MS/MS analysis of isolated nucleoli shows enrichment for nucleolar proteins.
DOI: https://doi.org/10.7554/eLife.45205.009
**Figure supplement 2.** LC-MS/MS-based identification of significantly changed proteins in the nucleolus upon heat shock.
DOI: https://doi.org/10.7554/eLife.45205.010
**Figure supplement 3.** HS induces nucleolar accumulation of EZH2 in leukemic cells and primary human peripheral blood stem cells.
DOI: https://doi.org/10.7554/eLife.45205.011
**Figure supplement 4.** Identification of changes in the nucleolar proteome in untreated and heat shocked K562 GFP-CBX8 cells.
DOI: https://doi.org/10.7554/eLife.45205.012

consistent with observations using model proteins, which are targeted to the nucleolus for post-stress degradation (*Park et al., 2013*).

Independent LC-MS/MS analysis with K562 GFP-CBX8 cells confirmed these findings, implying that HS-induced nucleolar accumulation of various chromatin regulators, protein chaperones and proteasomal subunits is a conserved biological phenomenon (*Figure 3—figure supplement 4A–E*, *Supplementary file 3*). Our data suggests that upon HS many chromatin remodelers and transcriptional regulators accumulate in the nucleolus which may become a hot spot for protein quality control.

## HS alters polycomb complex binding and the epigenetic state at target genes

In *Drosophila*, nucleolar accumulation of Pc has been suggested to contribute to the generalized silencing of most of the genome observed in heat shocked cells (*Li et al., 2015*). Combined with our observations that many chromatin remodeling and transcription regulatory proteins accumulate in the nucleolus after HS, these data prompted us to speculate that such HS-induced redistributions may severely impact on the chromatin bound fraction of various epigenetic regulators. To analyze changes in PRC1 complex chromatin binding upon HS we performed ChIP-qPCRs in K562 cells expressing PcG GFP-fusion proteins and validated PRC1 binding to PcG target genes. Indeed, upon HS (1 hr, 44°C) we observed a strong reduction in chromatin binding of GFP-CBX2, BMI1-GFP, MEL18-GFP and, to a lesser extent, GFP-RING1B (*Figure 4A*). To investigate how HS impacts on chromatin binding of endogenous PRC1 subunits, we performed ChIPs using an antibody directed against endogenous CBX8, and also here we observed a quick reduction in CBX8 binding to Polycomb target genes after HS (*Figure 4B*). Moreover, in line with our LC-MS/MS data, ChIP analysis using an antibody directed against the PRC2 subunit EZH2 also clearly showed a loss of endogenous EZH2 binding from target genes after HS, confirming that both PRC1 and PRC2 show strongly reduced chromatin binding after HS (*Figure 4C*).

To investigate how this may functionally impact on PRC1/2-deposited epigenetic marks, we analyzed H2AK119ub and H3K27me3 levels at PcG target genes. Previous studies have shown that HS induces a rapid but reversible loss of ubiquitinated histones (*Carlson et al., 1987*; *Mimnaugh et al., 1997*). Indeed, we also observed a strong decrease in H2AK119ub levels after HS (*Figure 4D*). In addition, H3K27me3 levels were also significantly reduced in heat-shocked cells vs. control cells (*Figure 4E*). These data show that HS not only causes displacement of PRC1/2 complexes but also leads to a reduction of their respective epimarks. To assay HS-induced loss of PRC1 chromatin binding in a genome-wide manner, we performed ChIP-seq of both endogenous CBX8 (K562 cells) and GFP-CBX2 (K562 GFP-CBX2 cells) in untreated cells or after HS. Heat maps and band plots of CBX8

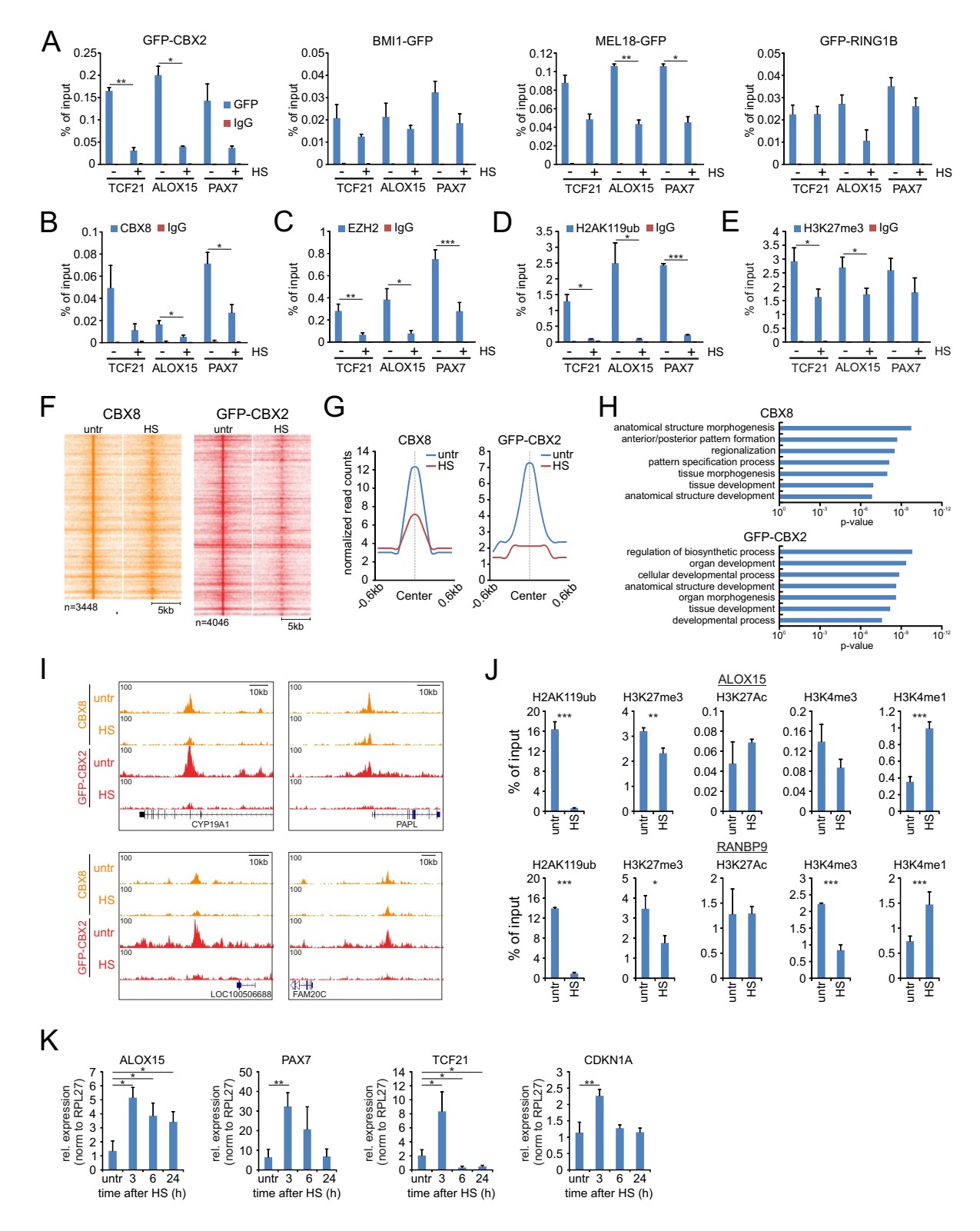

**Figure 4.** Heat shock induces loss of PRC1 and PRC2 binding to target genes and changes in epigenetic marks. (**A**) ChIP-qPCR analyses of GFP-CBX2, BMI1-GFP, MEL18-GFP and GFP-RING1B binding to Polycomb target genes in untreated and heat shocked (1 hr, 44°C) cells. ChIP reactions were performed using an anti-GFP antibody on cells expressing the respective GFP-fusion protein. Error bars represent mean ±range (n = 2, independent replicates, statistical analysis was performed using Student's t-test, *p<0.05 and **p<0.01). (**B**) ChIP-qPCR analyses of Polycomb target genes using an

*Figure 4 continued on next page*

*Figure 4 continued*

antibody directed against endogenous CBX8 in untreated and heat shocked (1 hr, 44˚C) K562 cells. Error bars represent mean ±range (n = 2 independent replicates, *p<0.05). (C) ChIP-qPCR analyses of Polycomb target genes using an antibody directed against endogenous EZH2 in untreated and heat shocked (1 hr, 44˚C) K562 cells. Error bars represent mean ± SD (n = 4 independent replicates, *p<0.05, **p<0.01 and ***p<0.001). (D) ChIP-qPCR analyses of H2AK119ub levels at Polycomb target genes in untreated and heat shocked (1 hr, 44˚C) K562 cells. Error bars represent mean ± SD (n = 3 independent replicates, *p<0.05 and ***p<0.001). (E) ChIP-qPCR analyses of H3K27me3 levels at Polycomb target genes in untreated and heat shocked (1 hr, 44˚C) K562 cells. Error bars represent mean ± SD (n = 3 independent replicates, *p<0.05). (F) ChIP-seq heatmap of endogenous CBX8 peaks (K562) and GFP-CBX2 peaks (K562 GFP-CBX2) and surrounding regions (−5 to + 5 kb) as identified in untreated cells and the respective signal in heat shocked cells (1 hr, 44˚C). (G) Band plots showing the median CBX8 and GFP-CBX2 signal (relative read counts) in untreated and heat shocked cells. (H) GO analyses of genes associated with endogenous CBX8 of GFP-CBX2 peaks in untreated cells show enrichment for developmental processes. (I) Characteristic examples of loci that show reduced binding of CBX8 and GFP-CBX2 upon HS. (J) ChIP-qPCR analyses of H2AK119ub, H3K27me3, H3K27Ac, H3K4me3 and H3K4me1 levels in untreated and heat shocked (1 hr, 44˚C) K562 cells. Error bars represent mean ± SD (technical triplicates, *p<0.05, **p<0.01 and ***p<0.001). qPCR analyses of the expression of Polycomb target genes in wild type K562 cells (untreated and in time after HS [1 hr, 44˚C]). Error bars represent mean ± SD (n = 3 independent replicates, *p<0.05 and **p<0.01).

DOI: https://doi.org/10.7554/eLife.45205.013

The following figure supplement is available for figure 4:

**Figure supplement 1.** CBX8 and GFP-CBX2 genome-wide peak localization analysis and CBX8 and GFP-CBX2 ChIP-seq signals at TSS-associated peaks.

DOI: https://doi.org/10.7554/eLife.45205.014

and GFP-CBX2 peaks clearly showed a loss of CBX8 and GFP-CBX2 chromatin binding after HS (*Figure 4F and G*, *Supplementary files 4–5*). In line with our previous observations concerning PRC1 chromatin occupancy, many CBX8 and GFP-CBX2 peaks were found to be intergenic (*van den Boom et al., 2016*) (*Figure 4—figure supplement 1A*). To investigate HS-induced reduction of CBX8 and GFP-CBX2 at gene promoters, we generated heat maps and band plots of transcription start site (TSS)-associated peaks (*Figure 4—figure supplement 1B–C*). Clearly, also here a loss of CBX8 and GFP-CBX2 chromatin association is observed upon HS. TSS-localized peaks are associated to genes enriched for development-related GO terms, confirming that these are Polycomb target genes (*Figure 4H*). Typical examples of chromatin regions that show a reduction of CBX8 or GFP-CBX2 chromatin binding are shown in *Figure 4I*. Next, we analyzed HS-induced changes of other epigenetic modifications. Here we find that, in addition to loss of H2AK119ub and H3K27me3, after HS (1 hr, 44˚C) also H3K4me3 levels are reduced, and, likely as a consequence of H3K4me3 loss, H3K4me1 levels are increased (*Figure 4J*). The increase in H3K4me1 levels argues that loss of epimarks is not a mere consequence of decreased nucleosome occupancy but truly a consequence of changes in the epigenetic marking of the chromatin. Finally, we investigated whether the HS-induced loss of PRC1/2-associated epimarks also resulted in loss of silencing of Polycomb target genes. Indeed, we found that the expression of various Polycomb target genes was increased 3 hr after HS, and recovered to pre-HS levels afterwards (*Figure 4K*). Taken together, these data show that, in parallel to the reallocation of chromatin remodelers and transcriptional regulators to the nucleolus, HS induces a quick reduction in PRC1 and PRC2 chromatin binding and changes in the epigenetic profile of PcG target genes with consequences for the transcriptional state of these genes.

## Altered GFP-CBX2 protein dynamics in the nucleolus upon HS

Next, we aimed to determine the physical-dynamic properties of PcG in the nucleoli of heat shocked cells. To achieve this, we stably expressed GFP-CBX2 in HeLa cells and determined GFP-CBX2 dynamics between the nucleolus and the nucleoplasm by photobleaching GFP-CBX2 in the nucleoli and analyzing the fluorescent recovery after photobleaching (FRAP) in time (*Figure 5A*). Strikingly, whereas the kinetics of nucleolar GFP-CBX2 in untreated cells were very quick, nucleolar GFP-CBX2 was highly immobile in heat shocked cells (*Figure 5B and C*) with limited to no dynamic exchange between the nucleolus and the nucleoplasm.

The nucleolus is a membrane-less nuclear body with liquid-like properties; its formation depends on liquid-liquid phase transition (*Brangwynne et al., 2011*; *Marko, 2012*). Analysis of the intranucleolar dynamics of GFP-CBX2 in heat shocked cells using FRAP/FLIP (fluorescence loss in photobleaching) analysis (*Figure 5A*), revealed that there was very little fluorescence recovery in the bleached region and fluorescence loss in the adjacent nucleolar region (*Figure 5D and E*), indicating

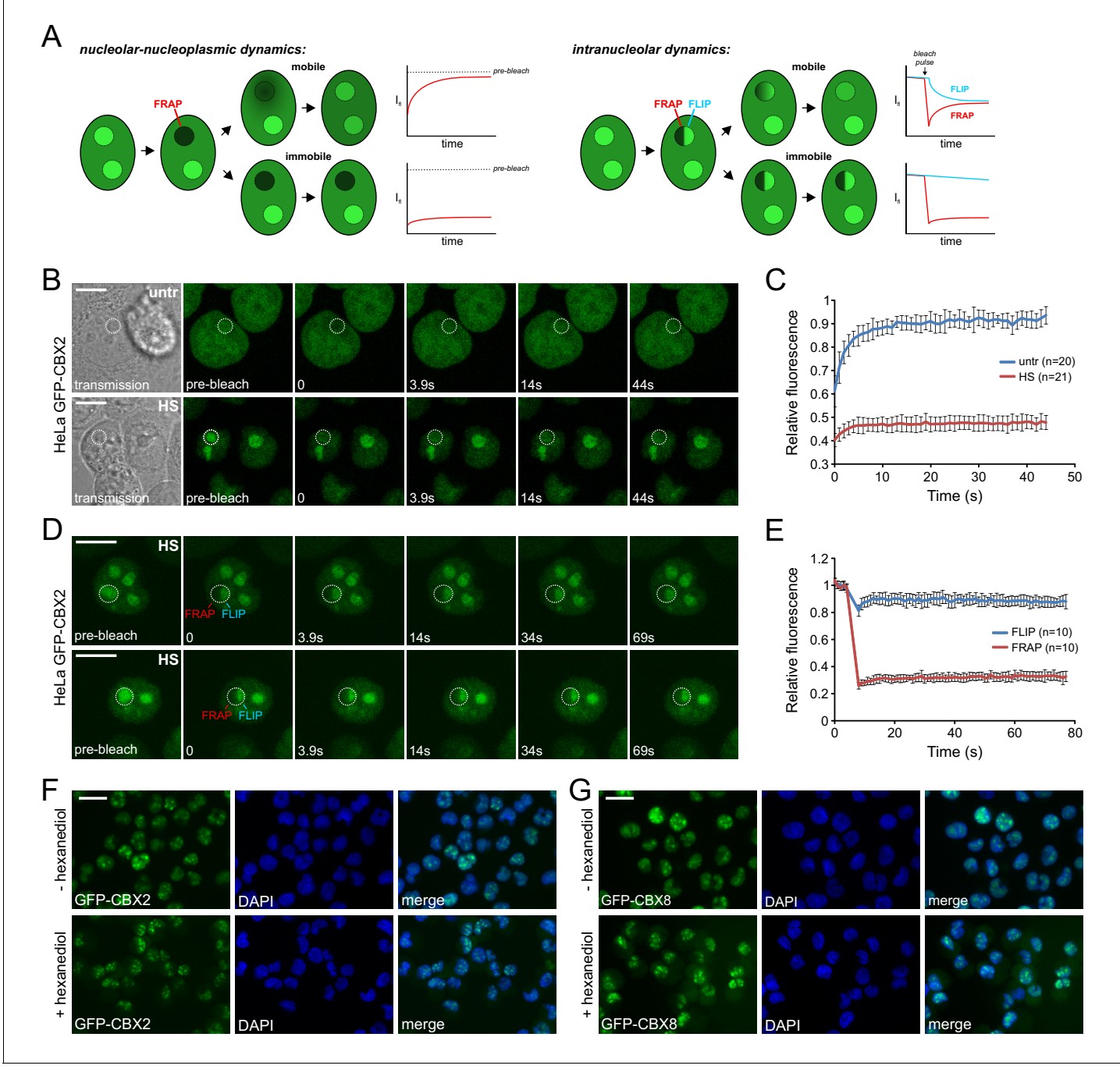

**Figure 5.** Heat shock strongly immobilizes GFP-CBX2 in the nucleolus in a 1,6-hexanediol insensitive manner. (**A**) Graphical summary of photobleaching experiments. (**B**) Representative example of Spot-FRAP analysis on HeLa GFP-CBX2 cells that were either untreated or heat shocked (30 min, 44°C). Confocal analysis was performed at 37°C directly after HS. FRAP region is indicated in the pre-bleach and transmission image. Scale bar represents 10 µm. (**C**) Average FRAP signals in the bleached nucleolar areas, starting directly after photobleaching. Error bars indicate mean ± SD. (**D**) Two representative examples of FLIP/FRAP analyses within the nucleolus of HeLa GFP-CBX2 cells that were either untreated or heat shocked (30 min, 44°C). FRAP and FLIP regions are indicated in the pre-bleach images. Scale bar represents 10 µm. (**E**) Average FLIP and FRAP signals in the nucleolus, starting before photobleaching. Error bars indicate mean ± SD. (**F**) GFP-CBX2 localization in K562 GFP-CBX2 cells that were heat shocked (1 hr, 44°C) and subsequently cultured at 37°C for 1 hr in the presence or absence of 10% 1,6-hexanediol. Scale bar represents 25 µm. (**G**) GFP-CBX8 localization in K562 GFP-CBX8 cells that were heat shocked (1 hr, 44°C) and subsequently cultured at 37°C for 1 hr in the presence or absence of 10% 1,6-hexanediol. Scale bar represents 25 µm.

DOI: https://doi.org/10.7554/eLife.45205.015

that GFP-CBX2 might be present in these nucleoli in a more solid-like state. Proteins in similar solid states have been shown to be aggregated (*Patel et al., 2015*). To discriminate between a liquid-like or solid state of CBX protein accumulations, we exposed heat shocked K562 GFP-CBX2 and K562 GFP-CBX8 cells to 1,6-hexanediol, an aliphatic alcohol that disturbs weak hydrophobic interactions (*Kroschwald et al., 2015*; *Patel et al., 2007*). Clearly, both GFP-CBX2 and GFP-CBX8 nucleolar accumulations were 1,6-hexanediol insensitive (*Figure 5F and G*), in line with a more solid, aggregation-like state.

## Heat-shock proteins modulate CBX protein recovery

We next asked whether the HS-induced allocation of PcG proteins to nucleoli serves to allow for a quick recovery of the epigenetic modifiers to restore chromatin binding of epigenetic regulators and associated changes in the epigenetic landscape after heat shock. In addition, we argued that the co-appearance of HSPs in these nucleoli may be required to recover these regulators from their solid-like, aggregated state. Certain mammalian HSPs have been demonstrated to be able to disentangle protein aggregates (*Mogk et al., 2018*; *Nillegoda et al., 2018*), including the HSPs (DNAJB1 and HSP70) that we identified as part of the nucleolar proteome after heat shock (*Figure 3H*). To test the reversibility of nucleolar GFP-CBX2 accumulations after HS, we treated K562 GFP-CBX2 cells with a HS (30 min, 44°C) and monitored GFP-CBX2 localization at 37°C afterwards. Clearly, within 3 hr after HS the GFP-CBX2 nucleolar accumulations dispersed and GFP-CBX2 regained its original pan-nuclear distribution (*Figure 6A*), showing that nucleolar accumulations of PcG proteins are readily reversible. Similarly, and in line with our cellular fraction data and LC-MS/MS data, we also observed a reversible HS-induced nucleolar accumulation of DNAJB1 (*Figure 6—figure supplement 1A*). In addition, also HSP70 showed post-HS nuclear translocation, and localized to the nucleolus in a reversible fashion (*Figure 6—figure supplement 1B*). Accumulation of HSP70 in the nucleolus was not as prominent as observed for DNAJB1, which is in line with our LC-MS/MS data, and may be a consequence of other nuclear activities of HSP70. Next, we investigated whether drug-induced inhibition of the HSP70 machinery using the HSP70 inhibitor VER-155008 would impact on the relocalization of PcG proteins to the nucleolus or would delay recovery of GFP-CBX2 from the nucleolus after HS. Clearly, post-HS nucleolar accumulation was not impaired upon HSP70 inhibition, suggesting that the HSP70 machinery is not involved in chaperoning PcG proteins to the nucleolus after HS (*Figure 6B*). However, HSP70 inhibition led to a clear delay in recovery of GFP-CBX2 from the nucleoli suggesting that HSP70 activity is required to resolve the nucleolar accumulation of GFP-CBX2 (*Figure 6B and C*). Similarly, a partial knockdown of HSPA1A, an abundant heat-inducible HSP70 family member in human cells and identical to HSPA1B at the protein level, resulted in a significant delay of GFP-CBX2 recovery from the nucleoli in HEK293T GFP-CBX2 cells (*Figure 6—figure supplement 1C–E*). Next, we investigated whether induction of endogenous HSPs accelerates GFP-CBX2 recovery after HS. K562 GFP-CBX2 cells were treated with two consecutive HSs with a 3-hr interval (*Figure 6D*). HS is known to induce HSP expression (including several HSP70s), resulting in a period of increased thermotolerance. This cellular property allowed us to compare the kinetics of GFP-CBX2 recovery in the presence of basal or HS-induced HSP levels. Indeed, whereas recovery of GFP-CBX2 after the first HS required almost 3 hr, recovery after the second HS was almost completed within 30 min (*Figure 6E and F*). Importantly, the amount of GFP-CBX2 that initially accumulated during the first or second HS did not change dramatically (*Figure 6E and F*). This implies that increased HSP levels specifically affect the recovery of GFP-CBX2 proteins from the solid phase within the nucleolus. Next, we tested whether the kinetics of recovery of GFP-CBX2 from the nucleolus and epigenetic recovery after HS were similar. Indeed, we found that H2AK119ub was completely recovered at 4 hr after HS, whereas H3K27me3 recovery was slightly delayed (*Figure 6G*). At later time points after HS we observed a complete recovery of GFP-CBX2 and various epigenetic modifications (*Figure 6—figure supplement 2A*). Interestingly, H2AK119ub recovery was strongly dependent on HSP protein expression. Thermotolerant cells that received a second HS displayed a much faster recovery of H2AK119ub compared to cells after the first HS (*Figure 6H*). In line with our ChIP-qPCR data, H3K27me3 was also reduced albeit with slower kinetics. Both HSP70 and DNAJB1 were strongly induced upon the first HS, whereas EZH2 and CBX8 protein levels were rather stable or slightly reduced. Next, we investigated whether post-HS H2AK119ub recovery is dependent on Polycomb proteins that are recovered from the nucleolus and not merely on newly translated Polycomb proteins. Therefore, we pre-treated K562 cells for 1 hr with cycloheximide or DMSO and studied

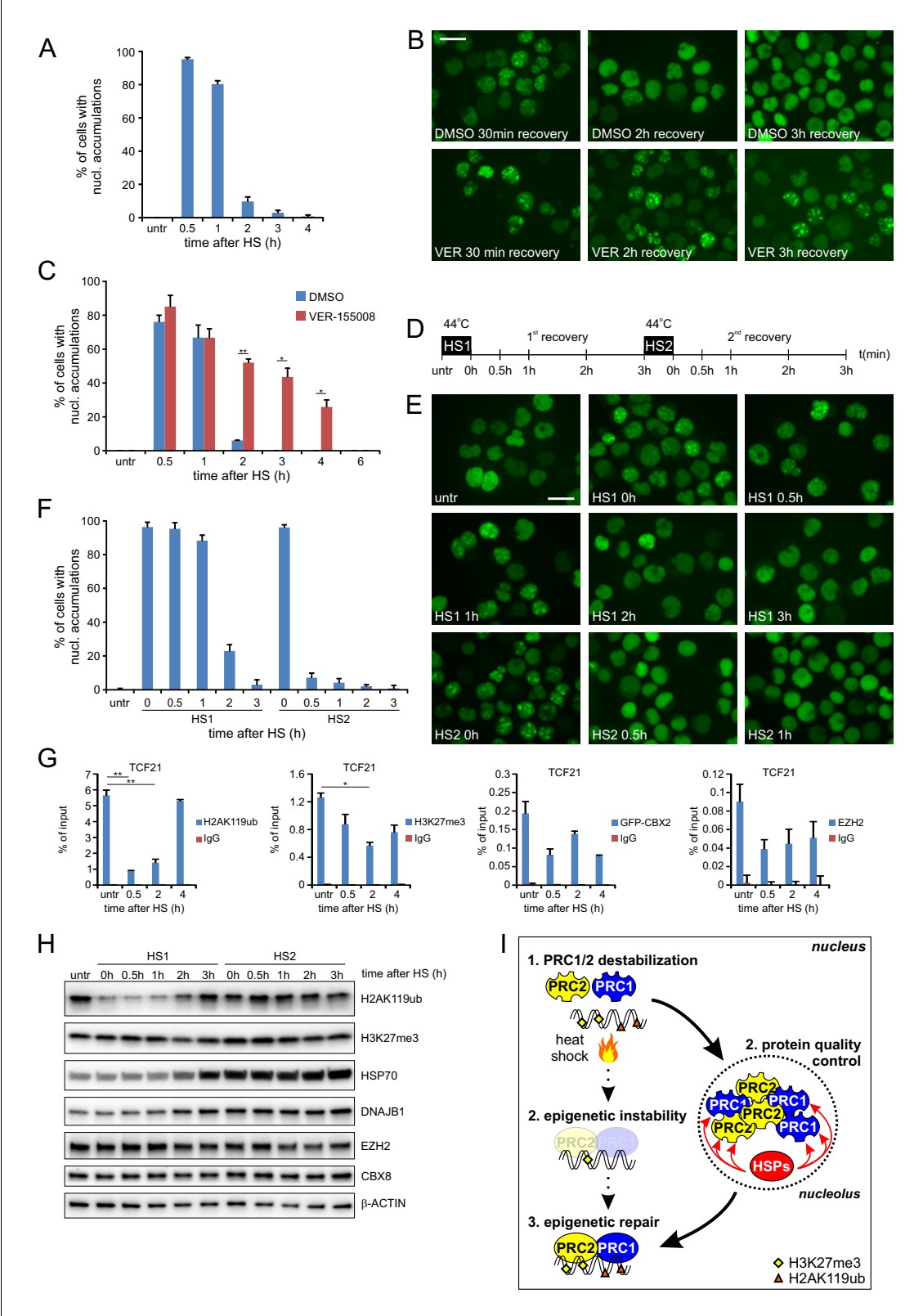

**Figure 6.** Post-HS nucleolar recovery of CBX proteins and epigenetic recovery depends on heat-shock protein activity. (**A**) Percentage of cells with nucleolar accumulation of GFP-CBX2 during recovery at 37˚C after HS (30 min, 44˚C). Error bars indicate the mean ± range calculated from independent microscopical images (n = 2; total cell number 100–170) (**B**) Representative images of fixed K562 GFP-CBX2 cultured at 37˚C after HS (30 min, 44˚C) for indicated time intervals in the presence of 5 µM VER-155088 or DMSO. Scale bar represents 25 µm. (**C**) Percentage of K562 GFP-CBX2 cells with

*Figure 6 continued on next page*

*Figure 6 continued*

nucleolar accumulations. Cells are cultured at 37°C after HS (30 min, 44°C) in the presence of 5 µM VER-155088 or DMSO. Error bars indicate the mean ± range calculated from independent microscopical images (n = 2; total cell number 50–90). Statistical analysis was performed using Student's t-test, *p<0.05, **p<0.01. Similar results were obtained in independent experiments. (D) Experimental design of thermotolerance experiment. (E) Representative images of K562 GFP-CBX2 cells fixed at indicated time points according to panel D. Scale bar represents 25 µm. (F) Quantification of percentage of K562 GFP-CBX2 cells with nucleolar accumulations at time points according to panel D. Error bars indicate the mean ± SD calculated from independent microscopical images (n = 5; total cell number 230–350). (G) ChIP-qPCR analyses of H2AK119ub, H3K27me3, GFP-CBX2 and endogenous EZH2 levels at the TCF21 locus in K562 GFP-CBX2 cells, either untreated or cross-linked at indicated time-points after a heat shock (30 min, 44°C). Error bars represent mean ± range (n = 2, independent replicates, *p<0.05 and **p<0.01). (H) Western blot analysis of H2AK119ub, H3K27me3, HSP70, DNAJB1, EZH2, CBX8 and β-ACTIN levels in K562 GFP-CBX2 cells samples isolated at indicated time points according to panel D. (I) Schematic representation of the effects of heat shock on PRC1/2 chromatin binding and epigenetic marks. Protein quality control in the nucleolus leads to refolding of Polycomb proteins, resulting in reinstallation of epigenetic modifications at Polycomb target genes.

DOI: https://doi.org/10.7554/eLife.45205.016

The following figure supplements are available for figure 6:

**Figure supplement 1.** GFP-CBX2, DNAJB1 and HSP70 show comparable HS-induced relocalization kinetics and post-HS GFP-CBX2 recovery is delayed upon HSP70 knockdown.
DOI: https://doi.org/10.7554/eLife.45205.017

**Figure supplement 2.** HS-induced epigenetic changes are reversible and H2AK119ub recovery is not dependent on de novo protein synthesis.
DOI: https://doi.org/10.7554/eLife.45205.018

H2AK119ub recovery after HS. Importantly, H2AK119ub recovery did not depend on de novo protein synthesis as H2AK119ub levels also recovered in cycloheximide-treated cells (*Figure 6—figure supplement 2B*). Taken together, these data show that molecular chaperones are crucial for recovery of GFP-CBX2 from the nucleolus and that this activity is essential for epigenetic recovery after HS (*Figure 6I*).

## Discussion

Spatial separation of proteins in the cytosol and nucleus upon heat stress has been repeatedly suggested to prevent interference of un- or misfolded proteins with essential cellular processes (*Escusa-Toret et al., 2013*; *Kaganovich et al., 2008*; *Miller et al., 2015*). At the same time, this 'storage' may allow for rapid recovery of proteins to re-initiate the crucial processes they are normally engaged in. For the nucleolus, these experiments have been done with reporter proteins, without any direct connection to physiological cellular processes. Despite this drawback, these studies revealed that HS-induced redistribution to the nucleolus is important for both refolding (*Nollen et al., 2001*) or degradation (*Park et al., 2013*) of these reporters. Our data are the first to show that numerous endogenous chromatin regulators temporarily accumulate in the nucleolus upon a HS, and, in an HSP70-dependent manner, fully and functionally recover to the chromatin upon return to physiological temperatures.

At this stage, we do not know what drives the association of the various PcG proteins to the nucleolus upon HS. Given the protein denaturation effects of HS, (partial) protein unfolding is likely a key driver of protein relocalization to the nucleolus. Whether it is unfolding of nucleolar proteins that cause retention of PcG proteins or actually partial unfolding of PcG proteins (or both) remains to be elucidated. Data from Audas and colleagues showed that a ncRNA, arising from intergenic stretches in between ribosomal DNA repeats, is capable of sequestering and immobilizing various proteins in the nucleolus upon acidosis or HS (*Audas et al., 2012*). Alternatively or in parallel, heat-unfolded nuclear reporter proteins (*Nollen et al., 2001*) or cytosolic proteins (*Park et al., 2013*) have been reported to accumulate in the nucleolus via chaperoned transport. Whilst we cannot exclude this possibility here, the appearance of PcG proteins in the nucleolus were independent of the HSP70 machinery. However, like in other studies (*Nollen et al., 2001*; *Welch and Feramisco, 1984*), we did find significant enrichment of HSP70 family members, and co-chaperones such as DNAJB1 and HSPB1, in the nucleolus after HS. We also show that both HSPA1A/HSP70 knockdown and inhibiting HSP70 activity result in a significant delay of GFP-CBX2 recovery from the nucleolus, whereas elevated HSP70 expression accelerates the reallocation of PcG proteins to the chromatin.

Whereas the nucleolus is a membrane-less organelle with liquid-like properties (*Brangwynne et al., 2011*; *Marko, 2012*), our finding that post-HS GFP-CBX2 accumulations are 1,6-hexanediol insensitive, suggest that it is present in the nucleolus in a more solid phase. This could relate to the requirement of an active HSP70 machinery for its re-solubilization upon recovery. In fact, both HSP70 and DNAJB1 that accumulate in the nucleolus are crucial components of chaperone machines with protein disaggregation power capable of functionally solubilizing proteins (*Mogk et al., 2018*; *Nillegoda et al., 2018*). It is important to note that the functional recovery of epigenetic control was not dependent on de novo synthesis of the PcG proteins. Translation inhibition did not interfere with this H2AK119ub recovery, suggesting that a least a large fraction of nucleolar PcG proteins are re-solubilized and functionally refolded. In addition, a fraction of the nucleolar accumulated proteins may be targeted for proteasomal degradation supported by the HS-induced nucleolar enrichment of the 26S proteasome that we found. In line with these data, a link between the proteasome and the nucleolus was previously suggested and proteasome inhibition leads to nucleolar accumulation of the proteasome (*Arabi et al., 2003*; *Fátyol and Grummt, 2008*; *Latonen et al., 2011*).

Interestingly, also other types of stress such as transcription inhibition, DNA damage induction and viral infection, have been shown to cause major changes in the protein composition of the nucleolus (*Andersen et al., 2005*; *Boisvert et al., 2010*; *Emmott et al., 2010*; *Lam et al., 2010*). However, whereas HS mainly induced accumulation of proteins in the nucleolus, transcription inhibition using actinomycin D resulted in a release of ribosomal proteins and RNA processing factors and an increase of snRNP proteins (*Andersen et al., 2005*). In contrast, treatment of cells with the proteasome inhibitor MG132, which similarly to HS leads to proteotoxic stress, led to an increase in ribosomal proteins in the nucleolus (*Andersen et al., 2005*). Together, these data suggest that the nucleolus could be an important protein quality control center serving under many different stress conditions.

Proteotoxic stress-induced loss of H2AK119ub has previously been observed by other groups (*Carlson and Rechsteiner, 1987*; *Dantuma et al., 2006*; *Mimnaugh et al., 1997*), and it has been proposed that the reason for this loss is the urgent need for 'free' ubiquitin in cells post-HS. Our data suggests that HS-induced redistribution of PcG proteins to the nucleolus has direct implications for histone marking and is not limited to H2AK119ub, but also affects H3K27me3, H3K4me3 and H3K4me1 levels at PcG target genes. Other studies have also shown HS-induced loss of H3K27Ac from HS-repressed enhancers (*Chen et al., 2017*) and changes in the 3D chromatin structure and epigenetic landscape in *Drosophila* cells (*Li et al., 2015*). In this latter study, the authors observed a moderate localization of the *Drosophila* Polycomb (Pc) protein to the nucleolus upon HS. Although ribosomal DNA transcription in the nucleolus was strongly reduced upon HS, Pc binding to ribosomal DNA repeats was not increased, suggesting that Pc is not involved in repressing ribosomal DNA transcription (*Li et al., 2015*). Whereas our HS-induced CBX protein accumulation in the nucleolus is more robust, we did not observe an increase in rDNA binding by CBX8 (data not shown). We also did not find increased H3K27me3 and H2AK119ub levels in the nucleolus after HS suggesting that accumulating PRC1 and PRC2 subunits are functionally inactive. Together these findings contradict a hypothetical chromatin regulatory activity of PRC1/2 in the nucleolus after HS but rather suggest that their respective subunits are undergoing protein quality control.

Taken together, our data shows that HS directly affects chromatin binding of PcG proteins and results in a decrease of PcG-related epigenetic modifications. Importantly, HSP70-dependent protein disaggregation and refolding enables PcG proteins to quickly re-initiate their epigenetic functions at target genes. It is evident that quick re-installation of PcG epimarks is key to maintain proper epigenetic regulation of PcG target genes. The question remains how often cells will erroneously 'repair' the epigenetic profile after stress-induced epigenetic instability. Despite the fact that the majority of epimarks may be properly reinstalled, mistakes will result in epigenetic scars, which may contribute to cellular transformation, loss of cell function and ultimately cell death. Various studies have shown that alterations in transcriptional and epigenetic regulation are major contributors to aging-associated loss of cellular function (*Booth and Brunet, 2016*). Based on our data, it is tempting to speculate that cellular-stress-induced epigenetic changes may contribute to age-associated epigenetic alterations. Importantly, it has been shown that the molecular chaperone system of a cell displays age-associated functional decline (*Klaips et al., 2018*; *Labbadia and Morimoto, 2015*).

This may well trigger age- or disease-associated reduction in protein quality control of epigenetic regulators, including PcG proteins, leading to alterations in the epigenome.

We propose a model where HS leads to loss of chromatin binding and nucleolar accumulation of PcG proteins and various other epigenetic regulators, likely as a consequence of protein unfolding. Loss of PcG chromatin binding leads to a loss of PRC1/2-related epigenetic modifications which recovery depends on HSP70 activity in the nucleolus.

# Materials and methods

## Key resources table

| Reagent type (species) or resource | Designation | Source or reference | Identifiers | Additional information |
|---|---|---|---|---|
| Cell line (*H. sapiens*) | K562 | ATCC | CCL-243 RRID:CVCL_0004 | |
| Cell line (*H. sapiens*) | HL60 | ATCC | CCL-240 RRID:CVCL_0002 | |
| Cell line (*H. sapiens*) | HeLa | ATCC | CCL-2 RRID:CVCL_0030 | |
| Cell line (*H. sapiens*) | HEK293T | ATCC | CRL-3216 RRID:CVCL_0063 | |
| Transfected construct (*H. sapiens*) | pRRL SFFV GFP-CBX2 | *van den Boom et al., 2016*; PMID: 26748712 | | Lentivirally transduced in K562, HeLa and HEK293T, stable cell lines |
| Transfected construct (*H. sapiens*) | PC182 GFP-CBX4 | *Vandamme et al. (2011)*; PMID: 21282530 | | Retrovirally transduced, stable cell line |
| Transfected construct (*H. sapiens*) | PC182 GFP-CBX6 | *Vandamme et al. (2011)*; PMID: 21282530 | | Retrovirally transduced, stable cell line |
| Transfected construct (*H. sapiens*) | PC182 GFP-CBX7 | *Vandamme et al. (2011)*; PMID: 21282530 | | Retrovirally transduced, stable cell line |
| Transfected construct (*H. sapiens*) | PC182 GFP-CBX8 | *Vandamme et al. (2011)*; PMID: 21282530 | | Retrovirally transduced, stable cell line |
| Transfected construct (*H. sapiens*) | pRRL SFFV GFP-CBX2 (aa2-63) | This study | | Lentivirally transduced, stable cell line |
| Transfected construct (*H. sapiens*) | pRRL SFFV GFP-CBX2 (aa2-96) | This study | | Lentivirally transduced, stable cell line |
| Transfected construct (*H. sapiens*) | pRRL SFFV GFP-CBX8 (aa2-62) | This study | | Lentivirally transduced, stable cell line |
| Transfected construct (*H. sapiens*) | pRRL SFFV GFP-CBX8 (aa2-96) | This study | | Lentivirally transduced, stable cell line |
| Antibody | GFP | Abcam | Cat# ab290 RRID:AB_303395 | WB (1:1000); ChIP 2 μg |
| Antibody | EZH2 | Cell Signalling Technology | Cat# 5246 (D2C9) RRID:AB_2797901 | WB (1:1000);IF (1:200); ChIP 5 μg |
| Antibody | SUZ12 | Abcam | Cat# ab12073 RRID:AB_442939 | WB (1:1000) |
| Antibody | CBX4 | Merck | Cat# 09–029 RRID:AB_1977084 | WB (1:1000) |

*Continued on next page*

*Continued*

| Reagent type (species) or resource | Designation | Source or reference | Identifiers | Additional information |
|---|---|---|---|---|
| Antibody | CBX4 | Cell Signalling Technology | Cat# 30559 (E6L7X) RRID:AB_2798991 | IF (1:100) |
| Antibody | CBX8 | Diagenode | Cat# C15410333 RRID:AB_2801424 | ChIP 2 µg |
| Antibody | CBX8 | Cell Signalling Technology | Cat# 14696 (D2O8C) RRID:AB_2687589 | WB (1:1000); IF (1:100) |
| Antibody | BMI1 | Merck | Cat# 05–637 RRID:AB_309865 | WB (1:1000) |
| Antibody | RING1B | Abcam | Cat# ab181140 (EPR12245) RRID:AB_2801425 | WB (1:1000) |
| Antibody | Fibrillarin | Abcam | Cat# ab5821 RRID:AB_2105785 | WB (1:1000), IF (1:100) |
| Antibody | NPM1 | Thermo Fisher Scientific | Cat# 32–5200 (FC-61991) RRID:AB_2533084 | IF (1:500) |
| Antibody | DNAJB1 | Enzo Life Sciences | Cat# ADI-SPA-450 RRID:AB_10621843 | IF (1:100) |
| Antibody | DNAJB1 | Enzo Life Sciences | Cat# ADI-SPA-400 RRID:AB_10631418 | WB (1:5000) |
| Antibody | HSP70 | Enzo Life Sciences | Cat# ADI-SPA-810 RRID:AB_10616513 | WB (1:5000); IF (1:100) |
| Antibody | HSP70 | StressMarq | Cat# C92F3A-5 RRID:AB_2570713 | WB (1:1000) |
| Antibody | H3K4me3 | Diagenode | Cat# C15410003 RRID:AB_2616052 | ChIP 2 µg |
| Antibody | H3K4me1 | Diagenode | Cat# C15410194 RRID:AB_2637078 | ChIP 2 µg |
| Antibody | H3K27me3 | Diagenode | Cat# C15410195 RRID:AB_2753161 | WB (1:1000); ChIP 2 µg |
| Antibody | H3K27Ac | Diagenode | Cat# C15410196 RRID:AB_2637079 | ChIP 2 µg |
| Antibody | H2AK119ub | Cell Signalling Technology | Cat# 8240 (D27C4) RRID:AB_10891618 | WB (1:1000); ChIP 2 µg |
| Antibody | H3 | Abcam | Cat# ab1791 RRID:AB_302613 | WB (1:1000) |
| Antibody | β-Actin | Santa Cruz Biotechnology | Cat# sc-47778 (C4) RRID:AB_2714189 | WB (1:1000) |
| Antibody | GAPDH | Fitzgerald Industries International | Cat# 10R-G109A RRID:AB_1285808 | WB (1:3000) |
| Antibody | Alexa Fluor 647 goat-anti-rabbit | Thermo Fisher Scientific | Cat# A-21244 RRID:AB_2535812 | IF (1:1000) |
| Antibody | Alexa Fluor 647 goat-anti-mouse | Thermo Fisher Scientific | Cat# A-21235 RRID:AB_2535804 | IF (1:1000) |
| Antibody | Alexa Fluor 488 goat-anti-rabbit | Thermo Fisher Scientific | Cat# A-11008 RRID:AB_143165 | IF (1:1000) |
| Antibody | Alexa Fluor 488 goat-anti-mouse | Thermo Fisher Scientific | Cat# A-11001 RRID:AB_2534069 | IF (1:1000) |
| Antibody | Goat Anti-Rabbit Immunoglobulins/ HRP | Agilent Dako | Cat# P044801-2 RRID:AB_2617138 | WB (1:5000) |

*Continued on next page*

*Continued*

| Reagent type (species) or resource | Designation | Source or reference | Identifiers | Additional information |
|---|---|---|---|---|
| Antibody | Rabbit Anti-Mouse Immunoglobulins/ HRP | Agilent Dako | Cat# P026002-2 RRID:AB_2801427 | WB (1:5000) |
| Sequence-based reagent | HSPA1A siRNAs | Dharmacon | Cat# M-005168–01 | |
| Chemical compound, drug | VER-155008 | Sigma-Aldrich | Cat# SML0271 | |
| Software, algorithm | GraphPad Prism | GraphPad Prism (https://graphpad.com) | RRID:SCR_015807 | Version 7.02 |
| Software, algorithm | ImageJ | ImageJ (http://imagej.nih.gov/ij/) | RRID:SCR_003070 | |
| Software, algorithm | MaxQuant | MaxQuant (http://www.biochem.mpg.de/5111795/maxquant) | RRID:SCR_014485 | Version 1.5.2.8 |
| Software, algorithm | Perseus | Perseus (http://www.perseus-framework.org) | RRID:SCR_015753 | Version 1.5.8.5 |

## Primary cell isolation

Cord blood (CB) and mobilized peripheral blood stem cells (mPBSCs) were obtained from healthy full-term pregnancies and allogeneic blood stem cell donors respectively after informed consent in accordance with the Declaration of Helsinki at the obstetrics departments at the Martini Hospital and University Medical Center Groningen. This study was approved by the UMCG Medical Ethical Committee. CB CD34$^+$ cells were isolated as previously described (*Schuringa et al., 2004*).

## Lenti- and retroviral transductions

CB CD34$^+$ cells, K562, HeLa, and HEK293T cells were transduced as described previously (*Horton et al., 2013*; *Schuringa et al., 2004*; *van den Boom et al., 2013*). One round of lentiviral transduction was performed and cells were harvested at day 2 after transduction. For retroviral trans-ductions virus was produced transiently in HEK293T cells by transfection of the appropriate PC182 GFP-fusion vector and pCL-Ampho at day 1. At day 2 the medium on the HEK293T cells was changed to RPMI (incl. 10% FCS and 1% P/S) and at day 3 the supernatant was harvested, filtered and used for infection of cells. To generate stable cell lines GFP-positive cells were sorted out 3 days after transduction.

## Cell culture

The (human) erythromyeloblastoid leukemia cell line K562 and HL60 cells were cultured in RPMI 1640 (containing L-glutamine) supplemented with 10% FCS and 1% penicillin/streptomycin (PAA Laboratories). CB CD34$^+$ cells were cultured in IMDM, supplemented with 20% FCS, 1% penicillin/ streptomycin, 20 ng/ml SCF, and 20 ng/ml IL-3. HeLa cells and HEK293T cells were cultured in DMEM supplemented with 10% FCS and 1% penicillin/streptomycin. Cell lines were all tested myco-plasma-free using a PCR-based assay. For VER-155008 treatment cells were pre-treated with VER-155008 at a concentration of 5 μM for 48 hr. Cycloheximide treatments were performed at a concen-tration of 10 μg/ml.

## GFP-fusion constructs

The lentiviral pRRL SFFV GFP-fusion vector for CBX2 was generated as described previously (*van den Boom et al., 2013*). Other GFP-CBX fusion proteins (PC182 GFP-CBX4, GFP-CBX6, GFP-CBX7 and GFP-CBX8) were expressed from retroviral vectors that were previously described (*Vandamme et al., 2011*). pRRL SFFV GFP-CBX2 (aa2-63), pRRL SFFV GFP-CBX2 (aa2-96), pRRL SFFV GFP-CBX8 (aa2-62), and pRRL SFFV GFP-CBX8 (aa2-96) were generated by PCR amplification

of the indicated fragment of the CBX2/8 protein using pRRL SFFV GFP-CBX2 and PC182 GFP-CBX8 as templates, followed by subcloning into the pJet1.2/blunt vector (Thermofisher). After sequence validation, these fragments were excised using BsrGI and subcloned into pRRL SFFV GFP-CBX2 where CBX2 was excised using BsrGI.

## Immunofluorescence microscopy

For immunofluorescence microscopy, cells were cytospinned on glass slides and subsequently fixed using 4% paraformaldehyde in PBS. Subsequently, cells were permeabilized using PBS containing 0.1% Triton X-100. Primary antibodies include anti-Fibrillarin (ab5821, Abcam), anti-NPM1 (FC-61991, Thermo Fisher Scientific), anti-CBX4 (E6L7X, Cell Signalling Technology) and anti-EZH2 (D2C9, Cell Signalling Technology) followed by secondary antibody staining using Alexa Fluor 488 goat-anti-rabbit (Thermo Fisher Scientific, A-11008), Alexa Fluor 488 goat-anti-mouse (Thermo Fisher Scientific, A-11001), Alexa Fluor 647 goat-anti-rabbit (Thermo Fisher Scientific, A-21244), or Alexa Fluor 647 goat-anti-mouse (Thermo Fisher Scientific, A-21235). Images were acquired on a Leica DM6000B microscope using a 40x dry objective (HCX PL FLUOTAR, numerical aperture: 0.75) or a 63x immersion objective (PL S-APO, numerical aperture: 1.30) using LAS-AF software (Leica). Confocal images were acquired on a Leica TCS SP8 confocal laser-scanning microscope using a HC PL APO CS2 63x/1.4 oil objective, and excitation with 488 nm (20 mW) and 633 nm (30 mW) laser lines.

## FRAP/FLIP analysis

For FRAP experiments, a confocal laser-scanning microscope (Zeiss LSM780 NLO; Carl Zeiss Microcopy) was used. HeLa GFP-CBX2 cells were seeded in a 35 mm dishes, no. 15 coverslip, 14 mm diameter (MatTek). To measure protein mobility after HS, cells were first heat shocked (30 min, 44° C), followed by FRAP analysis at 37°C. To perform FRAP experiments first the subnuclear location of nucleoli was identified using a transmission image. Next, a region of interest covering the whole nucleolus (in the case of FRAP experiments), or half the nucleolus (FRAP/FLIP) was defined. For FRAP/FLIP experiments the opposite side of the nucleolus was selected to measure the FLIP signal. The FRAP region was bleached for five iterations at the highest intensity of the 488 nm line of a 25 mW argon laser focused by a EC Plan-Neofluar 40x/1.30 Oil DIC M27 lens (Carl Zeiss Microcopy). Recovery of fluorescence was monitored at 1 s intervals at 0.5% of the laser intensity used for bleaching. For generation of FRAP and FLIP curves the background signal was subtracted from the measured fluorescent intensities and subsequently normalized to prebleach levels. Finally, the mean and standard deviation were plotted.

## siRNA-mediated HSP70 knockdown

For HSPA1A/HSP70 knockdown, HEK293T GFP-CBX2 cells were seeded on poly-L-lysine coated coverslips, and the next day mock or HSPA1A/HSP70 siRNAs were transfected using Lipofectamine 2000 (Thermo Fisher Scientific). Two days after transfection, cells were heat shocked and fixed at the indicated time points.

## Western blotting

Western blot analysis was performed as published previously (van den Boom et al., 2013). The following antibodies were used: anti-GFP (ab290, Abcam), anti-EZH2 (D2C9, Cell Signaling Technology), anti-SUZ12, anti-CBX4 (09–029, Merck), anti-CBX8 (C15410333, Diagenode), anti-RING1B (ab181140, Abcam), anti-BMI1 (F6, Merck), anti-DNAJB1 (SPA-400, Enzo Life Sciences), anti-HSP70 (SPA-810, Enzo Life Sciences), anti-Fibrillarin (ab5821, Abcam), anti-H3K27me3 (07–449, Merck), anti-H2AK119ub (D27C4, Cell Signaling Technology), anti-H3K4me3 (ab8580, Abcam), and anti-β-Actin (C4, Santa Cruz).

## Cellular fractionation and nucleoli isolation

Cellular fractionation and nucleoli isolation was essentially performed as described (Andersen et al., 2002) and http://www.lamondlab.com). Briefly, $1 \times 10^8$ K562 or HL60 cells were spun down and washed using PBS. Subsequently, cell pellets were resuspended in 2 ml ice-cold buffer A (10 mM Hepes pH 7.9, 1.5 mM $MgCl_2$, 10 mM KCl, 0.5 mM DTT, and protease inhibitors. Cells were incubated on ice for five minutes and subsequently broken open using a 2 ml dounce homogenizer (10

strokes using a tight pestle). Dounced cells were spun down at 4°C and the supernatant (cytoplasmic fraction) was stored. The pellet was resuspended in 1.2 ml buffer S1 (0.25 M Sucrose, 10 mM MgCl2, and protease inhibitors) and layered on 1.2 ml buffer S2 (0.35M Sucrose, 0.5 mM MgCl2, and protease inhibitors). The nuclei were spun though the sucrose cushion for 5 min at 1475 x g. Next, the nuclei were resuspended in 1.2 ml buffer S2 and sonicated 6 × 10 s on ice using a probe sonicator (Soniprep 150, MSE). The sonicated nuclei were layered on 1.2 ml buffer S3 (0.88M Sucrose, 0.5 mM MgCl2, and protease inhibitors) and centrifuged for 10 min at 2889 x g. The supernatant (nucleoplasmic fraction) was stored and the pellet was washed using 0.5 ml buffer S2 and finally resuspended in 80 microliter buffer S2. Subsequently, 20 microliter 5x Laemmli sample buffer was added and samples were boiled for 5 min.

## In gel trypsin digestion

Nucleoli samples were loaded on a 4–12% pre-cast NuPAGE gel (Invitrogen), and shortly ran into the gel. Gel staining was performed using Coomassie dye R-250 (Thermo Scientific) followed by destaining with ultrapure water. Coomassie-stained samples were excised in one gel slice that were further cut into small pieces and destained using 70% 50 mM $NH_4HCO_3$ and 30% acetonitrile. Reduction was performed using 10 mM DTT dissolved in 50 mM $NH_4HCO_3$ for 30 min at 55°C. Next the samples were alkylated using 55 mM iodoacetamide in 50 mM $NH_4HCO_3$ for 30 min at room temperature and protected from light. Subsequently, samples were washed for 10 min with 50 mM $NH_4HCO_3$ and for 30 min with 100% acetonitrile. Remaining fluid was removed and gel pieces were dried for 15 min. at 55°C. Tryptic digest was performed by addition of sequencing-grade modified trypsin (10 ng/µl in 50 mM $NH_4HCO_3$) and overnight incubation at 37°C. Peptides were extracted using 5% formic acid followed by a second elution with 5% formic acid in 75% acetonitrile. Samples were dried in a SpeedVac centrifuge and dissolved in 5% formic acid.

## LC-MS/MS analysis

Online chromatography of peptides was performed with an Ultimate 3000 nano-HPLC system (Thermo Fisher Scientific) coupled online to a Q-Exactive-Plus mass spectrometer with a NanoFlex source (Thermo Fisher Scientific) equipped with a stainless steel emitter. Tryptic digests were loaded onto a 5 mm × 300 µm i.d. trapping micro column packed with PepMAP100 5 µm particles (Dionex) in 0.1% FA at the flow rate of 20 µL/min. After loading and washing for 3 min, peptides were forward-flush eluted onto a 50 cm × 75 µm i.d. nanocolumn, packed with Acclaim C18 PepMAP100 2 µm particles (Dionex). The following mobile phase gradient was delivered at the flow rate of 300 nL/min: 3–50% of solvent B in 90 min; 50–80% B in 1 min; 80% B during 9 min, and back to 3% B in 1 min and held at 3% B for 19 min. Solvent A was 100:0 H2O/acetonitrile (v/v) with 0.1% formic acid and solvent B was 0:100 H2O/acetonitrile (v/v) with 0.1% formic acid. MS data were acquired using a data-dependent top-10 method dynamically choosing the most abundant not-yet-sequenced precursor ions from the survey scans (300–1650 Th) with a dynamic exclusion of 20 s. Sequencing was performed via higher energy collisional dissociation fragmentation with a target value of 1e5 ions determined with predictive automatic gain control. Isolation of precursors was performed with a window of 1.8 Da. Survey scans were acquired at a resolution of 70,000 at m/z 200. Resolution for HCD spectra was set to 17,500 at m/z 200 with a maximum ion injection time of 50 ms. Normalized collision energy was set at 28. Furthermore, the S-lens RF level was set at 60 and the capillary temperature was set at 250degr. C. Precursor ions with single, unassigned, or six and higher charge states were excluded from fragmentation selection.

## Data analysis

Raw mass spectrometry data were analyzed using MaxQuant version, 1.5.2.8 (*Cox and Mann, 2008*), using default settings and LFQ/iBAQ enabled, and searched against the Human Uniprot/Swissprot database (downloaded June 26, 2016, 20197 entries). The data was further processed using Perseus software, version 1.5.8.5 (*Tyanova et al., 2016*).

## RNA isolation and qPCR

Total RNA was isolated using the RNeasy Mini Kit (QIAGEN), and cDNA was generated using the iScript cDNA synthesis kit (Bio-Rad). For quantitative RT-PCR, cDNA was amplified using

SsoAdvanced SYBR Green Supermix (Bio-Rad) on a MyIQ thermocycler (Bio-Rad). Primer sequences can be found in *Supplementary file 6*.

## ChIP-qPCR

ChIP analysis was essentially performed as described previously (*Frank et al., 2001*). ChIP reactions were performed using the following antibodies: anti-GFP (ab290, Abcam), anti-CBX8 (C15410333, Diagenode), anti-EZH2 (D2C9, Cell Signaling), anti-H2AK119ub (D27C4, Cell Signaling), anti-H3K27me3 (C15410195, Diagenode), anti-H3K27Ac (C15410196, Diagenode), anti-H3K4me1 (C15410194, Diagenode), and anti-H3K4me3 (C15410003, Diagenode). ChIP efficiencies were assessed using qPCR. Primer sequences can be found in *Supplementary file 6*.

## ChIP-seq

Sequencing samples were prepared according to the manufacturer's protocol (Illumina). End repair was performed using the precipitated DNA using Klenow and T4 PNK. A 3' protruding A base was generated using Taq polymerase and adapters were ligated. The DNA was loaded on gel and a band corresponding to ~ 300 bp (ChIP fragment + adapters) was excised. The DNA was isolated, amplified by PCR and used for cluster generation on the Illumina NextSeq 500 genome analyzer. The 50 bp tags were mapped to the human genome HG19 using BWA (*Li and Durbin, 2009*). For processing and manipulation of SAM/BAM files SAMtools was used (*Li et al., 2009*). For each base pair in the genome, the number of overlapping sequence reads was determined and averaged over a 10 bp window and visualized in the UCSC genome browser (*Kent et al., 2002*).

## Detection of enriched regions

Peak calling algorithm MACS was used to detect the binding sites at a q-value cut off for peak detection of 0.01. ChIP-seq tracks were visualized using UCSC genome browser (*Kent et al., 2002*). Identification of genes associated to detected peaks was performed using GREAT (*McLean et al., 2010*). The accession number the ChIP-seq data in this paper is GEO: GSE121182.

## Tag counting

Tags within a given region were counted and adjusted to represent the number of tags within a 1 kb region. Subsequently, the percentage of these tags as a measure of the total number of sequenced tags of the sample was calculated.

## Generation of profiles and heatmaps

Heatmaps and bandplot profiles were generated using fluff (*Georgiou and van Heeringen, 2016*).

## GO analysis

Gene ontology (GO) analysis was performed using BiNGO (*Maere et al., 2005*).

# Acknowledgements

We thank Marcel de Vries (Interfaculty Mass Spectrometry Center, Groningen) for help with mass spectrometry analyses. This work is supported by a grant from the Dutch Cancer Foundation (RUG 2014–6832). Part of this work was performed at the UMCG Microscopy and Imaging Center (UMIC), which is sponsored by NWO-grant 175-010-2009-023.

# Additional information

### Funding

| Funder | Grant reference number | Author |
|---|---|---|
| KWF Kankerbestrijding | RUG 2014-6832 | Jan Jacob Schuringa<br>Vincent van den Boom |

The funders had no role in study design, data collection and interpretation, or the decision to submit the work for publication.

## Author contributions

Maria Azkanaz, Aida Rodríguez López, Bauke de Boer, Wouter Huiting, Investigation, Writing—review and editing; Pierre-Olivier Angrand, Edo Vellenga, Resources, Writing—review and editing; Harm H Kampinga, Methodology, Writing—review and editing; Steven Bergink, Joost HA Martens, Investigation, Methodology, Writing—review and editing; Jan Jacob Schuringa, Conceptualization, Supervision, Funding acquisition, Methodology, Writing—original draft, Writing—review and editing; Vincent van den Boom, Conceptualization, Supervision, Funding acquisition, Investigation, Methodology, Writing—original draft, Writing—review and editing

## Author ORCIDs

Jan Jacob Schuringa (iD) https://orcid.org/0000-0001-8452-8555
Vincent van den Boom (iD) https://orcid.org/0000-0002-1992-9608

## Decision letter and Author response

Decision letter https://doi.org/10.7554/eLife.45205.029
Author response https://doi.org/10.7554/eLife.45205.030

# Additional files

## Supplementary files

• Supplementary file 1. Label-free quantification of proteins detected in nucleoli from untreated and heat shocked K562 cells (exp 1). Table contains LC-MS/MS data, and label-free quantification data of identified proteins in nucleoli isolated from untreated and heat shocked K562 cells (experiment 1).
DOI: https://doi.org/10.7554/eLife.45205.019

• Supplementary file 2. Label-free quantification of proteins detected in nucleoli from untreated and heat shocked K562 cells (exp 2). Table contains LC-MS/MS data, and label-free quantification data of identified proteins in nucleoli isolated from untreated and heat shocked K562 cells (experiment 2).
DOI: https://doi.org/10.7554/eLife.45205.020

• Supplementary file 3. Label-free quantification of proteins detected in nucleoli from untreated and heat shocked K562 GFP-CBX8 cells. Table contains LC-MS/MS data, and label-free quantification data of identified proteins in nucleoli isolated from untreated and heat shocked K562 GFP-CBX8 cells.
DOI: https://doi.org/10.7554/eLife.45205.021

• Supplementary file 4. Endogenous CBX8 peaks detected in K562 cells. Table contains positional information of identified endogenous CBX8 peaks based on CBX8 ChIP-seq data in K562 cells.
DOI: https://doi.org/10.7554/eLife.45205.022

• Supplementary file 5. GFP-CBX2 peaks detected in K562 GFP-CBX2 cells. Table contains positional information of identified GFP-CBX2 peaks based on GFP-CBX2 ChIP-seq data in K562 GFP-CBX2 cells.
DOI: https://doi.org/10.7554/eLife.45205.023

• Supplementary file 6. Primer sequences. Table contains sequence information of all primers used for quantitative RT-PCR and ChIP-qPCR.
DOI: https://doi.org/10.7554/eLife.45205.024

• Transparent reporting form
DOI: https://doi.org/10.7554/eLife.45205.025

## Data availability

Numerical data of proteomics experiments can be found in Supplementary files 1-3. Additional data on detected peaks in our ChIP-seq data sets can be found in Supplementary files 4 and 5. Sequencing data have been deposited in GEO under accession codes GSE121182.

The following dataset was generated:

| Author(s) | Year | Dataset title | Dataset URL | Database and Identifier |
|---|---|---|---|---|
| Azkanaz M, Rodrí-guez López A, de Boer B, Huiting W, Angrand PO, Vel-lenga E, Kampinga HH, Bergink S, Martens JHA, Schuringa JJ, van den Boom V | 2019 | Protein quality control in the nucleolus safeguards recovery of epigenetic regulators after heat shock | https://www.ncbi.nlm.nih.gov/geo/query/acc.cgi?acc=GSE121182 | NCBI Gene Expression Omnibus, GSE121182 |

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
