## [Decision Letter]

Thank you for submitting your article "Protein quality control in the nucleolus safeguards recovery of epigenetic regulators after heat shock" for consideration by *eLife*. Your article has been reviewed by two peer reviewers, and the evaluation has been overseen by a Reviewing Editor and Kevin Struhl as the Senior Editor. The reviewers have opted to remain anonymous.

The reviewers have discussed the reviews with one another and the Reviewing Editor has drafted this decision to help you prepare a revised submission.

Summary:

The manuscript by Azkanaz et al. presents a thorough and convincing study on how Polycomb repressive complexes redistribute from chromatin to the nucleolus during heat shock. This redistribution is temporally coincident with reduced binding of polycomb proteins and the loss of epigenetic marks on H2AK119ub and H3K27me3. Loss of these marks is then correlates with the silencing of PRC1/2 target genes. Interestingly, while these Polycomb complexes are present within the nucleolus upon heat shock, FRAP/FLIP assays show the nucleolus acquires a more solid state versus its more fluidic state as seen under normal growth conditions. Upon recovery from heat shock, the PRC 1/2 and their associated CBX proteins leave the nucleolus, and the authors suggest that this is dependent on HSP70. Polycomb proteins associate again with chromatin, thus re-establishing their epigenetic marks. The authors suggest the nucleolus functions as a protein quality control center during times of stress (heat shock). The topic is timely, the manuscript is well written and figures are well presented. With several revisions (as outlined below), this manuscript should be suitable for publication.

Essential revisions:

1) The main, overarching concern is that this study is heavily relies on the overexpression of GFP-tagged proteins and thus it is not possible to establish which part of the fusion protein, or whether an artificial high level of expression, is responsible for the reported effects. This is partly tackled with experiments included in Figure 3, showing that endogenous Polycomb proteins accumulate in the nucleolus upon heat shock by proteomics. However, it would be required to perform additional immunofluorescence of endogenous Polycomb proteins to demonstrate nucleolar accumulation of endogenous proteins (at normal cellular levels) upon heat shock.

To accomplish this goal, the authors are requested to consider the following:

a) Given that most data rely on cell lines expressing GFP-CBX fusions, the authors should explain how they concluded that these proteins are fully functional. Functionality of at least the GFP-CBX2 fusion should be solidly established.

b) Experiments are carried out mainly in one leukemic cell line and thus it is not possible to rule out that nucleolar accumulation of Polycomb proteins is a cell type-specific phenomenon. Proof that endogenous Polycomb proteins accumulate in the nucleolus upon heat shock in a different cell model would increase the impact and value of the study. Primary non-transformed cells would be best.

c) The authors should conclusively demonstrate that the PRC1/2 and associated regulators are enriching within the granular components of nucleoli rather than in peri-nucleolar spaces. There is some confusion among the reviewers as to why DAPI should stain nucleoli in Figure 2D. Could this be chromatin that has accumulated in per-nucleolar spaces under stress conditions? Additional immunofluorescence using marker proteins for the granular component (e.g., Nucleostemin) should be performed.

2) In Figure 3D, are you saying SUZ12 rises in nucleolar abundance upon heat shock? Its nucleolar abundance seems rather modest. This should be clarified.

3) In many of the experiments, standard deviations were provided when n=2. When there are only 2 replicates, the range should be reported, not standard deviation. Please update all figures.

4) The authors claim that recovery of CBX proteins is dependent on HSP70. However, they present very weak lines of evidence to support this link. At least, authors should present confocal analysis to study colocalization of Polycomb proteins and HSP70 upon heat shock and during recovery. Also, proof of reduced reversibility could be presented in a genetic loss of function experiment.

5) There are also several places in the manuscript that were deemed unclear, and where editing and/or clarification is recommended.

a) In the Introduction (second paragraph) is the PRC1 complex dependent on the PRC2 complex? This portion of the Introduction was not too clear.

b) The final sentence of paragraph two in subsection “HS induces large-scale changes in the nucleolar proteome” should be re-written for improved clarity.

c) Subsection “HS alters Polycomb complex binding and the epigenetic state at target genes”, and Figure 4 legend: are these K562 cells?

d) Paragraph two of subsection “HS alters Polycomb complex binding and the epigenetic state at target genes” could be edited just for clarity: the loss of the PRC complexes from the chromatin during heat shock does not explain global genome silencing doing heat shock, correct?

e) In Figure 4J and paragraph two of subsection “HS alters Polycomb complex binding and the epigenetic state at target genes”, as H3K4me3 declines and H3K4me1 rises, can I assume H3K4me3 is losing two methyl groups to give rise to H3K4me1?

f) The first sentence of paragraph two in subsection “Altered GFP-CBX2 protein dynamics in the nucleolus upon HS” could be written: "The nucleolus is a membrane-less nuclear body with liquid-like properties; its formation depends…" Related to this, do the authors have any speculation on what makes nucleoli more solid upon heat shock? I thought this was very interesting.

g) The statement “Thermotolerant cells that received a second HS displayed a much faster recovery of H2AK119ub compared to cells after the first HS (Figure 6H). Importantly, H2AK119ub recovery did not depend on de novo protein synthesis as H2AK119ub levels also recovered in cycloheximide treated cells (Figure 6—figure supplement 1B)” in the Results were confusing until I read the third paragraph in the Discussion. Perhaps the authors could better explain in the Results why they blocked protein synthesis with cycloheximide and exactly when the drug was added with respect to heat shock.

---

## [Author Response]

Essential revisions:1) The main, overarching concern is that this study is heavily relies on the overexpression of GFP-tagged proteins and thus it is not possible to establish which part of the fusion protein, or whether an artificial high level of expression, is responsible for the reported effects. This is partly tackled with experiments included in Figure 3, showing that endogenous Polycomb proteins accumulate in the nucleolus upon heat shock by proteomics. However, it would be required to perform additional immunofluorescence of endogenous Polycomb proteins to demonstrate nucleolar accumulation of endogenous proteins (at normal cellular levels) upon heat shock.To accomplish this goal, the authors are requested to consider the following:a) Given that most data rely on cell lines expressing GFP-CBX fusions, the authors should explain how they concluded that these proteins are fully functional. Functionality of at least the GFP-CBX2 fusion should be solidly established.

This is indeed an important point raised by the reviewers. To address this, we have included new data in Figure 1—figure supplement 1 where we now show, by means of GFP pull-outs, that GFP-CBX2 is properly incorporated in the PRC1 complex (panel A), and that GFP-CBX2 target genes largely overlap with endogenous CBX2 target genes (based on ENCODE/Broad Institute data; Ram et al., 2011) and H2AK119ub enriched genes (our own data from Van den Boom et al., 2016). In Van den Boom et al. (2016), we also performed independent ChIP-seq analysis on K562 GFP-CBX2 cells and found that over 96% of the identified GFP-CBX2 target genes were also targeted by GFP-RING1B in K562 GFP-RING1B cells. Taken together, these data show that the GFP-tag does not impair the integration of GFP-CBX2 into the PRC1 complex and does not prevent GFP-CBX2 chromatin binding to sites that largely overlap with sites occupied by endogenous CBX2.

b) Experiments are carried out mainly in one leukemic cell line and thus it is not possible to rule out that nucleolar accumulation of Polycomb proteins is a cell type-specific phenomenon. Proof that endogenous Polycomb proteins accumulate in the nucleolus upon heat shock in a different cell model would increase the impact and value of the study. Primary non-transformed cells would be best.

To study accumulation of endogenous Polycomb proteins, we used antibodies directed against CBX4 and EZH2 that worked efficiently in fluorescence microscopy experiments. We also tried several anti-CBX2 antibodies but these did not yield good signals. Confocal analysis of endogenous CBX4 clearly displays HS-induced accumulation in the nucleolus, both in K562 cells (Figure 1H) and HL60 cells (leukemic cell line, Figure 1—figure supplement 2G), in line with data we observed for GFP-CBX4 (Figure 1F). We have also included confocal images showing HS-induced nucleolar accumulation of endogenous EZH2 in HL60 cells, K562 cells, and primary non-transformed CD34^+^ peripheral blood stem cells (PBSCs) (Figure 3—figure supplement 3A-C). Finally, we have added an additional cellular fractionation study where we isolated nucleoli from HL60 cells (Figure 2G and Figure 3D). Also in HL60 cells we observe a robust nucleolar accumulation of endogenous CBX4, CBX8, RING1B, EZH2 and SUZ12 upon HS.

Taken together, these data clearly show that endogenous Polycomb proteins also accumulate in nucleoli after HS and that this behavior is observed in multiple cell types, including HL60 cells and primary non-transformed CD34^+^ mobilized peripheral blood stem cells (mPBSCs).

c) The authors should conclusively demonstrate that the PRC1/2 and associated regulators are enriching within the granular components of nucleoli rather than in peri-nucleolar spaces. There is some confusion among the reviewers as to why DAPI should stain nucleoli in Figure 2D. Could this be chromatin that has accumulated in per-nucleolar spaces under stress conditions? Additional immunofluorescence using marker proteins for the granular component (e.g., Nucleostemin) should be performed.

To further specify the subnucleolar localization of Polycomb proteins upon HS we have now included NPM1 as a marker for the granular layer of nucleoli. Confocal images of K562 cells stained with NPM1 and Fibrillarin antibodies (Figure 1—figure supplement 2B) clearly shows that, as expected, NPM1 is localized around Fibrillarin in the nucleolus (untreated and heat shocked). Confocal and fluorescence microcopy analysis of heat shocked K562 GFP-CBX2 cells co-stained with NPM1 antibody shows that GFP-CBX2 is mostly colocalizing with NPM1 in the granular layer or is surrounded by the NPM1 staining (Figure 1D and Figure 1—figure supplement 2C-D), underlining the fact that GFP-CBX2 truly accumulates in the nucleolus upon heat shock.

Concerning the DAPI signal in the isolated nucleoli in Figure 2D: we acknowledge that the intra-nucleolar DAPI signal indeed is lower than in the surrounding chromatin, but still stains the nucleoli due to the presence of genomic DNA in the form of rDNA repeats. Due to the lack of intense extranucleolar DAPI signal this may now be picked up more easily in the nucleolar preparations. In line with our findings, in a recent study in *A. thaliana* cells, purified and FACS sorted nucleoli were stained DAPI positive (Pontvianne et al., 2016). Importantly, the intranucleolar DAPI signal is not dependent on a stress condition since the signal is equal to that of nucleoli isolated from untreated cells.

2) In Figure 3D, are you saying SUZ12 rises in nucleolar abundance upon heat shock? Its nucleolar abundance seems rather modest. This should be clarified.

Based on our LC-MS/MS data and Western blot analysis of cellular fractions, we indeed conclude that SUZ12 also enriches in the nucleoli of K562 cells. We have now included cellular fractionation data of untreated and heat shocked HL60 cells that show an even more robust accumulation of SUZ12 in the nucleolar fraction upon HS.

3) In many of the experiments, standard deviations were provided when n=2. When there are only 2 replicates, the range should be reported, not standard deviation. Please update all figures.

We have adjusted the figures and legends accordingly.

4) The authors claim that recovery of CBX proteins is dependent on HSP70. However, they present very weak lines of evidence to support this link. At least, authors should present confocal analysis to study colocalization of Polycomb proteins and HSP70 upon heat shock and during recovery. Also, proof of reduced reversibility could be presented in a genetic loss of function experiment.

This is also an important point made by the reviewers. We have now included confocal images of K562 GFP-CBX2 cells (untreated, directly after HS, 3h after HS) stained with antibodies directed against either DNAJB1 or HSP70 (Figure 6—figure supplement 1A-B). These experiments clearly show a nucleolar accumulation of DNAJB1 after HS, which is lost from the nucleoli in time after HS. Importantly, cells that show a recovery of GFP-CBX2 from the nucleoli 3h after HS also do not display DNAJB1 accumulation in the nucleolus anymore, suggesting that the kinetics of nuclear recovery for both proteins are very similar. Also for HSP70, which is excluded from the nucleoli in untreated cells, we observe a HS-induced nuclear translocation and a loss of nucleolar exclusion. Upon recovery of GPF-CBX2 from of the nucleoli in time after HS, we also find that HSP70 is again excluded from the nucleoli. The nucleolar accumulation of DNAJB1 is more prominent than that of HSP70, which is in line with our LC-MS/MS data, and may be a consequence of the requirement of HSP70 in other nuclear processes.

To test whether in an HSP70 loss-of-function setting nucleolar recovery of GFP-CBX2 is slowed down, we have stably expressed GFP-CBX2 in HEK293T cells and transfected these with mock siRNAs or HSPA1A/HSP70 siRNAs (Figure 6—figure supplement 1C-E). Clearly, in cells depleted of HSP70, a significantly delay in recovery of GFP-CBX2 from the nucleolus was seen.

Taken together, our previous and newly added data now show that after HS (i) HSP proteins accumulate in the nucleolus, (ii) the kinetics of HSP70/DNAJB1 recovery of nucleolar accumulation after HS resemble that of GFP-CBX2, (iii) HSP70 inhibition and partial HSPA1A/HSP70 knockdown both delay GFP-CBX2 nucleolar recovery, and (iv) HS-induced upregulation of HSP expression (thermal tolerance) accelerates GFP-CBX2 recovery.

Importantly, HSPA1A is the most abundant HSP70 family member expressed in human cells. However, since the HSPA1A and HSPA1B gene products are identical at the protein level, we cannot distinguish between both proteins in our LC-MS/MS analysis. Therefore, for reasons of clarity, we have changed in our Volcano plots HSPA1B to HSPA1A/B (Figure 3H, Figure 3—figure supplement 2E, Figure 3—figure supplement 4D). In the main body of the text we refer to these proteins as HSP70.

5) There are also several places in the manuscript that were deemed unclear, and where editing and/or clarification is recommended.

We edited our manuscript and clarified the text.

a) In the Introduction (second paragraph) is the PRC1 complex dependent on the PRC2 complex? This portion of the Introduction was not too clear.

This section has been rewritten and clarified.

b) The final sentence of paragraph two in subsection “HS induces large-scale changes in the nucleolar proteome” should be re-written for improved clarity.

This section has been rewritten and clarified.

c) Subsection “HS alters Polycomb complex binding and the epigenetic state at target genes”, and Figure 4 legend: are these K562 cells?

Yes, this is now explicitly mentioned in the text.

d) Paragraph two of subsection “HS alters Polycomb complex binding and the epigenetic state at target genes” could be edited just for clarity: the loss of the PRC complexes from the chromatin during heat shock does not explain global genome silencing doing heat shock, correct?

Indeed, loss of PRC1/2 complexes from the target genes does not induce gene silencing upon heat shock. We have now included expression data of a selected set of Polycomb target genes (Figure 4K), which shows that transcription of these genes is actually temporarily increased 3h after HS and recovered to normal at later time points.

e) In Figure 4J and paragraph two of subsection “HS alters Polycomb complex binding and the epigenetic state at target genes”, as H3K4me3 declines and H3K4me1 rises, can I assume H3K4me3 is losing two methyl groups to give rise to H3K4me1?

We indeed assume that increase of H3K4me1 is a direct consequence of loss of H3K4me3, we have made this more explicit in the text.

f) The first sentence of paragraph two in subsection “Altered GFP-CBX2 protein dynamics in the nucleolus upon HS” could be written: "The nucleolus is a membrane-less nuclear body with liquid-like properties; its formation depends…" Related to this, do the authors have any speculation on what makes nucleoli more solid upon heat shock? I thought this was very interesting.

We have adjusted the text accordingly. The immobilization of GFP-CBX2 in the nucleoli after HS is indeed fascinating. As we do not have any experimental data as to what is the cause of this immobilization, we have not included these speculations in the main text of the manuscript. However, it could be that partial unfolding of proteins causes them to have a strong increase in intranucleolar interaction with a very low dissociation constant. Whether this is due to a specific interaction with nucleolar components or as a consequence of the formation of molecular aggregates remains to be seen and is a focus for future studies.

g) The statement “Thermotolerant cells that received a second HS displayed a much faster recovery of H2AK119ub compared to cells after the first HS (Figure 6H). Importantly, H2AK119ub recovery did not depend on de novo protein synthesis as H2AK119ub levels also recovered in cycloheximide treated cells (Figure 6—figure supplement 1B)” in the Results were confusing until I read the third paragraph in the Discussion. Perhaps the authors could better explain in the Results why they blocked protein synthesis with cycloheximide and exactly when the drug was added with respect to heat shock.

We have changed the text accordingly and moved the rationale for this experiment to the Results section.

References:

Pontvianne F, Carpentier MC, Durut N2 Pavlištová V, Jaške K, Schořová Š, Parrinello H, Rohmer M, Pikaard CS, Fojtová M, Fajkus J, Sáez-Vásquez J. *Identification of Nucleolus-Associated Chromatin Domains Reveals a Role for the Nucleolus in 3D Organization of the A. thaliana Genome.* Cell Rep. 2016. 16(6):1574-1587. doi: 10.1016/j.celrep.2016.07.016.

Ram O1, Goren A, Amit I, Shoresh N, Yosef N, Ernst J, Kellis M, Gymrek M, Issner R, Coyne M, Durham T, Zhang X, Donaghey J, Epstein CB, Regev A, Bernstein BE. Combinatorial patterning of chromatin regulators uncovered by genome-wide location analysis in human cells. Cell. 2011 Dec 23;147(7):1628-39. doi: 10.1016/j.cell.2011.09.057.